# THE HIDDEN UNIFORM CLUSTER PRIOR IN SELF-SUPERVISED LEARNING

**Mahmoud Assran**[*,1,2,3], **Randall Balestriero**[1], **Quentin Duval**[1], **Florian Bordes**[1,3,4]
**Ishan Misra**[1], **Piotr Bojanowski**[1], **Pascal Vincent**[1,3,4], **Michael Rabbat**[1,3], **Nicolas Ballas**[1]

[1]Meta AI (FAIR)   [2]McGill University, ECE   [3]Mila, Quebec AI Institute   [4]Universite de Montreal, DIRO

## ABSTRACT

A successful paradigm in representation learning is to perform self-supervised pretraining using tasks based on mini-batch statistics (e.g., SimCLR, VICReg, SwAV, MSN). We show that in the formulation of all these methods is an overlooked prior to learn features that enable uniform clustering of the data. While this prior has led to remarkably semantic representations when pretraining on class-balanced data, such as ImageNet, we demonstrate that it can hamper performance when pretraining on class-imbalanced data. By moving away from conventional uniformity priors and instead preferring power-law distributed feature clusters, we show that one can improve the quality of the learned representations on real-world class-imbalanced datasets. To demonstrate this, we develop an extension of the Masked Siamese Networks (MSN) method to support the use of arbitrary features priors.

## 1 INTRODUCTION

Self-supervised pretraining has emerged as a highly effective strategy for unsupervised representation learning, with remarkable advances demonstrated by joint-embedding methods (Chen et al., 2020b; Caron et al., 2021; Bardes et al., 2021; Assran et al., 2022). In the context of visual data, these approaches typically learn representations by training a neural network encoder to produce similar embeddings for two or more views of the same image. However, since outputting a constant vector regardless of the input would satisfy this objective, one of the main challenges with joint-embedding methods is to prevent such pathological solutions. A common remedy is to employ a regularizer that maximizes the volume of space occupied by the representations. This is sometimes referred to as the volume maximization principle. In practice, the volume maximization principle is implemented in a variety of ways, for example, by contrasting negative samples (Bromley et al., 1993; He et al., 2019; Chen et al., 2020b), by removing correlations in the feature space (Bardes et al., 2021; Zbontar et al., 2021), or by finding high entropy clusterings of the data (Asano et al., 2019; Caron et al., 2020; Assran et al., 2021; 2022). When pretrained on the ImageNet dataset (Russakovsky et al., 2015), these methods have been shown to produce representations that encode highly semantic features (Caron et al., 2020; 2021; Assran et al., 2022).

However, the commonly used ImageNet-1K dataset is relatively class-balanced, which is in contrast to most real-world settings, where data is often *class-imbalanced* and semantic concepts follow a long-tailed power-law distribution (Newman, 2005; Mahajan et al., 2018; Van Horn et al., 2018). Indeed, it has been shown that pretraining the same joint-embedding methods on long-tailed datasets can lead to significant drops in performance (Tian et al., 2021a). Such an observation is problematic in that it significantly hinders the applicability of modern research advances with joint-embedding methods to real-world settings.

In this work, we explore the use of joint-embedding methods for class-imbalanced datasets. First, we theoretically show that current methods with volume maximization regularizers such as VICReg (Bardes et al., 2021), SwAV (Caron et al., 2020), MSN (Assran et al., 2022) and SimCLR (Chen et al., 2020b) (with limited assumptions), have a uniform feature prior; i.e., a bias to learn features that enable grouping the data into clusters of roughly equal size. Consequently, these joint-embedding

---

*massran@meta.com

methods will penalize features that do not uniformly cluster the data, even if such features correlate well with class information; see Figure 1.

Second, we empirically validate that joint-embedding methods employing volume maximization regularizers are sensitive to the mini-batch class distributions. These approaches fail to learn class-discriminative features when the samples within a mini-batch do not follow a uniform class distribution. This observation partially explains why performance degrades when pretraining with real-world data, where sampled mini-batches often contain highly imbalanced class distributions.

Finally, based on this observation, we propose to move away from conventional uniformity priors and instead reformulate self-supervised criteria to prefer long-tailed feature priors that are more aligned with the distribution of semantic concepts in real-world datasets. In particular, we extend Masked Siamese Networks (MSN) of Assran et al. (2022) to support the use of arbitrary features priors, and refer to this extension as *Prior Matching for Siamese Networks (PMSN)*. When pretraining on the iNaturalist 2018 dataset (Van Horn et al., 2018), which is naturally long-tailed, we demonstrate that moving away from uniform priors leads to more semantic representations and improved transfer on downstream tasks.

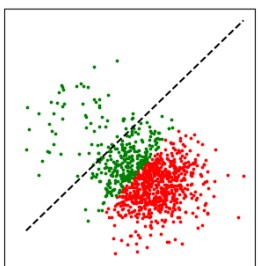

(a) K-means with class-balanced data

(b) K-means with class-imbalanced data

Figure 1: Impact of uniform cluster prior in K-means when class distribution of data is imbalanced. K-means clustering depicted in color (green vs red). Ground-truth cluster separation depicted with a dotted black line. When uniform feature prior is not satisfied, K-means can identify undesirable features for discriminating between data points.

## 2   BACKGROUND

Given the recent success of joint-embedding methods, there is a growing literature that aims to build a better understanding of their behaviour. Several works have sought to develop generalization bounds for joint-embedding methods with volume maximization penalties (Arora et al., 2019; Balestriero & LeCun, 2022). Other works have sought to better understand the differences between various volume maximization penalties and connect them under limited assumptions (Garrido et al., 2022). In general, it has been shown that $\ell_2$-normalized contrastive losses can be decomposed into an "alignment" plus volume maximization component that scatters the representations uniformly on the unit hypersphere (Wang & Isola, 2020). Following this observation, other works (Chen et al., 2021) have sought to reformulate contrastive losses to scatter representations either (a) uniformly on the unit hypercube, or (b) onto Gaussian distributions (which have the highest entropy amongst all distributions with a given variance). There is also theoretical work (Tian et al., 2021b) which aims to understand why certain joint-embedding methods, such as BYOL (Grill et al., 2020), can avoid representation collapse without explicit use of a volume maximization penalty.

While these works have helped build our understanding on the training dynamics of joint-embedding methods, they do not directly explain why empirical use of these methods with real-world class-imbalanced data has often led to a degradation in downstream task performance (Tian et al., 2021a; Goyal et al., 2022) (see Appendix A for a broader discussion of related work). In this work, we explore the use of joint-embedding methods with class-imbalanced data. In particular, we theoretically show that a broad range of methods (beyond contrastive) prescribe a uniform *feature* prior, and that this prior is detrimental when pretraining with class-imbalanced data.

## 3   UNIFORM PRIORS IN MODERN SELF-SUPERVISED LEARNING

In this section, we theoretically show that common SSL methods such a, VICReg (Bardes et al., 2021), SwAV (Caron et al., 2020), MSN (Assran et al., 2022), and (with limited assumptions) SimCLR (Chen et al., 2020b), correspond to variants of K-means, and thereby impose a uniform cluster prior; i.e., a bias to learn features that enable uniform clustering of the data. The governing assumption in K-means is the presence of isotropic data clusters, with roughly an equal number of

samples per cluster (Wu et al., 2009; Liang et al., 2012). When this assumption is not satisfied in practice, K-means may learn undesirable features for discriminating between samples (cf. Figure 1).

### 3.1 BACKGROUND: K-MEANS FORMULATIONS AND THE UNIFORM PRIOR

**Explicit (Centroid) K-means.** Recall that K-means proposes a centroid based clustering of the data. In particular, given a set of $N$ data points $\{\boldsymbol{x}_n\}_{n=1}^N$, K-means partitions the elements into $K$ disjoint groups $\mathbb{X}_1, \ldots, \mathbb{X}_K$, such that $\sum_{i=1}^K |\mathbb{X}_i| = N$. The K-means objective can be written as

$$\min_{\{\mathbb{X}_k\}_{k=1}^K} \sum_{k=1}^K \sum_{\boldsymbol{x} \in \mathbb{X}_k} \|\boldsymbol{x} - \mu_k\|_2^2, \tag{1}$$

where the optimization problem is to identify the members of the disjoint sets $\mathbb{X}_1, \ldots, \mathbb{X}_K$, and $\mu_k$ (the $k^{\text{th}}$ cluster centroid) is precisely the average of the members of $\mathbb{X}_k$.

**Implicit (Centroid) K-means.** One important note that we will carry through our study is that K-means does not require explicit computation of the cluster centroids $\mu_k$ to evaluate the objective. To see this standard result, one can relate the sum of pairwise distances to the sum of radial distances for any data partition,

$$\sum_{k=1}^K \sum_{\boldsymbol{x} \in \mathbb{X}_k} \|\boldsymbol{x} - \mu_k\|_2^2 = \sum_{k=1}^K \frac{1}{2|\mathbb{X}_k|} \sum_{\boldsymbol{x}, \boldsymbol{x}' \in \mathbb{X}_k} \|\boldsymbol{x} - \boldsymbol{x}'\|_2^2. \tag{2}$$

This result suggests that the K-means loss can be minimized by either learning cluster centroids (explicit K-means), or by learning cluster memberships (implicit K-means). We make this relation precise in Proposition 1, which is proven in Appendix C.1 and was already noted in Zha et al. (2001); Awasthi et al. (2015).

**Proposition 1.** *The explicit K-means problem, defined by learning a set of $K$ centroids $\mu_1, \ldots, \mu_K$, admits the same global optimum as the implicit K-means problem, defined by learning a cluster membership matrix $\boldsymbol{P} \in \{0, 1\}^{N \times K}$, such that $\boldsymbol{P}\mathbf{1}_K = \mathbf{1}_N$.*

The fundamental assumptions governing the success of K-means lie in having clusters with roughly the same number of samples and intra-cluster data covariance that is isotropic with the form $\sigma \boldsymbol{I}$, where $\sigma > 0$ and $\boldsymbol{I}$ is the identity matrix (Wu et al., 2009; Liang et al., 2012). In the sequel, we will show that various SSL methods can be seen as employing either explicit or implicit K-means.

### 3.2 HOW K-MEANS NATURALLY EMERGES FROM SELF-SUPERVISED LOSSES

In this section, we demonstrate how standard SSL methods naturally employ K-means at their core. Although this result might seem intuitive for methods that explicitly compute a clustering of the representations, we surprisingly found that this is also the case for some methods (e.g., VICReg) that do not explicitly involve centroid parameters or a clustering step.

**Implicit K-means: VICReg and SimCLR.** The VICReg loss (Bardes et al., 2021) consists of three terms, one measuring the $\ell_2$ distance between the positive view pairs, one encouraging the off-diagonal entries of the embedding covariance matrix to go to $0$, and one encouraging its diagonal entries to be greater than $1$. One standard simplification done in practice to study VICReg is to switch the variance and covariance terms to a single term so that the new VICReg loss is given by

$$\mathcal{L} = \alpha \|\text{Cov}(\boldsymbol{Z}) - \boldsymbol{I}\|_F^2 + \frac{\gamma}{N} \sum_{i,j=1}^N (\boldsymbol{G})_{i,j} \|\boldsymbol{z}_i - \boldsymbol{z}_j\|_2^2, \tag{3}$$

where $\text{Cov}(\boldsymbol{Z})$ is the covariance matrix of the vectors $\boldsymbol{z}_1, \ldots, \boldsymbol{z}_n$, the constants $\alpha, \gamma > 0$ are hyper-parameters, and $G_{i,j} \in \{0, 1\}$ is equal to 1 when the representations $z_i$ and $z_j$ correspond to positive views of the same image. From this formulation, we relate VICReg to the implicit K-means algorithm in Proposition 2, which is proven in Appendix C.2. In particular, note how the summation term on the right hand side of equation 3 recovers the right hand side of equation 2.

**Proposition 2.** *VICReg with hyper-parameters $\gamma \gg \alpha$ recovers the K-means loss from equation 1 on the embeddings, with an additional regularizer enforcing orthogonality of the centroids.*

The hyper-parameter requirement $\gamma \gg \alpha$ is commonly employed with VICReg in practice to ensure invariance to the pre-defined set of data-augmentations used to construct the positive image views. In fact, in the opposite case, where $\alpha \gg \gamma$, a degenerate whitening can be learned without keeping any information about the input samples due to the non-linearity of the deep neural network mapping, as shown in Balestriero & LeCun (2022). Note that under mild conditions, the VICReg loss has been shown to be equivalent to the SimCLR loss (Garrido et al., 2022). In such scenarios, Proposition 2 would also directly apply to the SimCLR method.

**Explicit K-means with soft constraints: MSN.** A common relaxation of constrained K-means is to remove the hard cluster assignment constraint, such that the condition $\boldsymbol{P}_{i,j} \in \{0, 1\}$ is generalized to $\boldsymbol{P}_{i,j} \in [0, 1]$ (Wang et al., 2010). This relaxes constrained K-means to the more general Gaussian Mixture Model (GMM) formulation, where each data point can partially belong to all clusters with some probability mass. Since, the derivation of the GMM loss (i.e., the ELBO and log-likelihood) is beyond the scope of this section, we simply state the objective as

$$\min_{\{\mu_k, \Sigma_k\}_{k=1}^K} \sum_{\boldsymbol{x} \in \mathbb{X}} \sum_{k=1}^{K} \frac{[p(\boldsymbol{x})]_k}{2} \|\boldsymbol{x} - \mu_k\|_{\Sigma^{-1}}^2 + N \sum_{k=1}^{K} \log \det(\Sigma_k) + \sum_{\boldsymbol{x} \in \mathbb{X}} D_{\mathrm{KL}}(p(\boldsymbol{x}) \| \boldsymbol{\pi}), \quad (4)$$

where $\boldsymbol{\pi}$ is the cluster prior and $[p(\boldsymbol{x})]_k$ is the posterior probability of input $\boldsymbol{x}$ belonging to cluster $k$, obtained from

$$p(\boldsymbol{x}) = \mathrm{softmax}\left(\boldsymbol{W}^T \boldsymbol{x} + \log(\boldsymbol{\pi}) - \frac{1}{2}\|\boldsymbol{x}\|_2^2 - \frac{1}{2}\mathrm{diag}(\boldsymbol{W}^T \boldsymbol{W})\right),$$

where $\boldsymbol{W} = [\mu_1, \dots, \mu_K]$ concatenates the centroids, and $\mathrm{diag}(\cdot)$ extracts the diagonal of its matrix argument into a column vector.

Of particular interest to us is the case in which the prior is set to the uniform distribution, $[\boldsymbol{\pi}]_k = 1/K$ for all $k \in [K]$, the covariance matrix is isotropic, $\Sigma_k = \sigma \boldsymbol{I}$ for $\sigma \geq 0$, and the centroids and data vectors are $\ell_2$-normalized, in which case equation 4 simplifies to

$$\min_{\mu_1, \dots, \mu_K} \sum_{\boldsymbol{x} \in \mathbb{X}} \sum_{k=1}^{K} \frac{[\mathrm{softmax}(\boldsymbol{W}^T \boldsymbol{x})]_k}{2} \|\boldsymbol{x} - \mu_k\|_2^2 - \sum_{\boldsymbol{x} \in \mathbb{X}} H\left(\mathrm{softmax}(\boldsymbol{W}^T \boldsymbol{x})\right). \quad (5)$$

One can also express the simplified GMM objective in equation 5 in terms of the marginal entropy (with a summation inside the entropy term) by following the ELBO derivations of Hoffman & Johnson (2016). The Masked Siamese Network (MSN) loss (Assran et al., 2022) with positive pairs $\boldsymbol{x}_n, \boldsymbol{x}_n^+$ and posteriors $\boldsymbol{p}_n = \mathrm{softmax}(\boldsymbol{W}^T \boldsymbol{x}_n / \sigma)$, $\boldsymbol{p}_n^+ = \mathrm{softmax}(\boldsymbol{W}^T \boldsymbol{x}_n^+ / \sigma)$, with temperature $\sigma$, is

$$\frac{1}{N} \sum_{n=1}^{N} H(\boldsymbol{p}_n^+, \boldsymbol{p}_n) - \lambda H(\overline{\boldsymbol{p}}), \quad (6)$$

where $\lambda > 0$, and $\overline{\boldsymbol{p}} \coloneqq \frac{1}{N} \sum_{n=1}^{N} \boldsymbol{p}_n$. We show that the MSN objective can be seen as variant of K-means variant employing an explicit cluster-membership penalty (cf. Appendix C.3).

**Proposition 3.** *MSN recovers the GMM loss from equation 5 with the variation that the $\ell_2$ distance is replaced with the cross-entropy distance of the posterior.*

**Explicit K-means with hard constraints: SwAV** Among popular K-means variants, one particular version requires specification of the cardinality of the clusters (Kleindessner et al., 2019; Bradley et al., 2000; Rujeerapaiboon et al., 2019). Specifically, the number of items in each cluster $|\mathbb{X}_k|$ is strictly enforced to take a specific value $N_k$. The resulting K-means formulation becomes

$$\min_{\{\mathbb{X}_k\}_{k=1}^K \text{s.t.} |\mathbb{X}_k| = N_k} \sum_{k=1}^{K} \sum_{\boldsymbol{x} \in \mathbb{X}_k} \|\boldsymbol{x} - \mu_k\|_2^2. \quad (7)$$

When $N_k = \frac{N}{K}$, we obtain strict enforcement of the uniform cluster prior, which is otherwise implicit in the formulation, but not strictly enforced. This variant of K-means was previously used in Wang et al. (2010). The SwAV (Caron et al., 2020) loss with positive posterior pairs $\boldsymbol{p}_n, \boldsymbol{p}_n^+$ (defined similarly to those in equation 6) is

$$\frac{1}{N} \sum_{n=1}^{N} H(\boldsymbol{p}_n^+, \boldsymbol{p}_n) \quad \text{subject to} \quad \boldsymbol{P}\mathbf{1}_N = \frac{N}{K}\mathbf{1}_K; \quad \boldsymbol{P}^\top \mathbf{1}_K = \mathbf{1}_N, \quad (8)$$

where $\boldsymbol{P} = [\boldsymbol{p}_1^+, \ldots, \boldsymbol{p}_n^+]$ concatenates the predictions. In Caron et al. (2020), this constraint is enforced in practice by projecting the matrix $\boldsymbol{P}$ onto the constraint set in each iteration using the Sinkhorn-Knopp algorithm (Cuturi, 2013). SwAV thus recovers a strictly constrained variant of K-means (cf. Appendix C.4).

**Proposition 4.** *SwAV recovers the constrained K-means loss from equation 5 with constraint enforcement through Sinkhorn-Knopp normalization.*

Note that for all presented SSL methods and their corresponding K-means variants, enforcing normalization of the centroids and features does not reduce the generality of the results; in this case, we simply obtain a correspondence to similar variants of spherical K-means (Hornik et al., 2012).

## 4 NEGATIVE EFFECT OF UNIFORM PRIORS FOR CLASS-IMBALANCED DATA

As proven in Section 3, joint-embedding methods employing volume maximization regularizers have an uniform feature prior. Following this observation, in this section we empirically demonstrate that such methods are sensitive to non-uniform mini-batch class distributions.

**Experimental setup.** We explore three joint-embedding methods employing diverse collapse prevention strategies: SimCLR (Chen et al., 2020b), VICReg (Bardes et al., 2021), and MSN (Assran et al., 2022). We also compare the performance of those models to instance-based methods such as MAE (He et al., 2021) and data2vec (Baevski et al., 2022), which do not employ volume maximization regularizers. In this evaluation, all models are pretrained on the ImageNet-1K dataset without access to the class labels. To explore sensitivity to mini-batch class distributions, we explore two dichotomous sampling strategies (which do require semantic knowledge of the image classes).

One strategy, termed *class-balanced sampling*, constructs the mini-batches in each iteration by first randomly selecting 960 classes out of the 1000 ImageNet-1K classes, and then sampling an equal number of images from each class.[1] Given that ImageNet is relatively class-balanced, this strategy produces mini-batches with similar statistics to traditional uniform sampling. Another strategy, termed *class-imbalanced sampling*, constructs the mini-batches in each iteration by first randomly selecting $K \ll 960$ classes out of the 1000 ImageNet-1K classes, and then sampling an equal number of images from each class, such that the total mini-batch size is the same as under the class-balanced sampling strategy. We pick $K$ as the smallest value needed to reach the default batch size for each respective method: 8 for SimCLR (which has the a default batch size of 4096) and 2 for VICReg and MSN (which have default batch sizes of 1024). Note that the class-imbalanced strategy maintains the same marginal probability of sampling individual data points as compared to the class-balanced sampling strategy (see Appendix G for a derivation of this equivalence).

After pretraining all models using the various sampling strategies, we evaluate performance on a wide range of downstream tasks requiring different levels of abstraction, i.e., classification with CIFAR100 (Krizhevsky et al., 2009), Places205 (Zhou et al., 2014), and iNat18 (Van Horn et al., 2018); object counting with Clevr/Count (Johnson et al., 2017); and depth prediction with Clevr/Dist (Johnson et al., 2017) and KITTI (Geiger et al., 2013). We also evaluate in-distribution performance of ImageNet classification (Russakovsky et al., 2015; Chen et al., 2020b). Additional pretraining and evaluation details can be found in Appendix D; the full set of results can be found in Appendix L.

**Empirical observations.** As can be seen in Table 1, the performance of joint-embedding methods employing volume maximization regularizers degrades significantly on all the semantic downstream tasks (IN1K, CIFAR100, Places205, Clevr/Count) when the mini-batches sampled during pretraining are not class-balanced (e.g., drops by as much as 17.7 top-1 on IN1K), but remain relatively stable (and even marginally improve) on low-level depth prediction tasks Clevr/Dist and KITTI, suggesting that class-imbalanced pretraining leads the model to capture lower-level (less semantic) features.

By contrast, evaluations with instanced-based methods data2vec and MAE in Table 1 show different trends. The MAE method employs a simple pixel-reconstruction loss for representation learning, and thus does not explicitly compute mini-batch statistics during pretraining. The data2vec method is more

---

[1]We choose 960 classes instead of 1000 so that the overall batch size is divisible by the number of GPUs utilized for distributed training.

Table 1: **Transfer:** Evaluation of the pretraining mini-batch sampling distribution on various downstream tasks. Each cell reports the task performance when pretraining with class-balanced sampling minus the task performance when pretraining with class-imbalanced sampling. Sampling imbalanced mini-batches during pretraining leads to a significant drop in image classification tasks for joint-embedding methods with volume maximization priors (SimCLR, MSN, VICReg), whereas instance-based methods, which do not employ such regularization (MAE, data2vec), are relatively unaffected.

|  | ImageNet | iNat18 | CIFAR100 | Places205 | Clevr/Count | Clevr/Dist | KITTI |
|---|---|---|---|---|---|---|---|
| SimCLR | -11.2 | -8.0 | -10.2 | -5.2 | -4.3 | +0.9 | +1.2 |
| MSN | -17.7 | -15.1 | -13.2 | -4.6 | -6.4 | +1.9 | -1.6 |
| VICReg | -17.7 | -17.3 | -12.0 | -6.0 | -3.0 | +0.7 | -1.1 |
| data2vec | -0.8 | +0.3 | -1.6 | +0.0 | -2.1 | -1.5 | -0.1 |
| MAE | -0.1 | +1.4 | +2.5 | +0.1 | -0.8 | +0.0 | +0.0 |

similar to MSN, SimCLR, and VICReg in that it utilizes a joint-embedding architecture; however, in contrast to those methods, data2vec does not explicitly compute mini-batch statistics during pretraining, and instead relies on architectural heuristics and careful hyperparameter choices to prevent collapse. When evaluating these methods with class-balanced pretraining versus class-imbalanced pretraining, we observe virtually no change in downstream task performance. Only methods with explicit volume maximization terms exhibit sensitivity to the mini-batch class distribution.

**Visualizing learned prototype vectors.** In Figure 2, we use RCDM (Bordes et al., 2022b) to visualize the prototypes learned by an MSN model pretrained on IN1K with either class-balanced or class-imbalanced mini-batch distributions. A prototype here refers to a row in the weight matrix of the final linear layer in the encoder. Each row in Figure 2 corresponds to samples generated by conditioning on a *single prototype* using various random seeds. Characteristics that remain constant across a row in Figure 2 reflect information contained in the prototype, whereas characteristics that vary reflect information that is not contained (i.e., to which the representations are invariant). When pretraining with class-balanced mini-batches, the emergent features tend to be associated with high-level concepts, such as specific ImageNet classes (Figure 2a). In contrast, when pretraining with class-imbalanced mini-batches, the learned features tend to be associated with low-level concepts, such as shape, pose, or texture (Figure 2b).

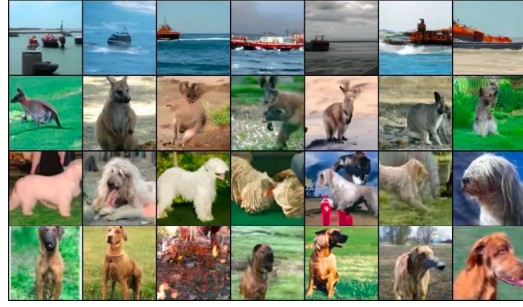 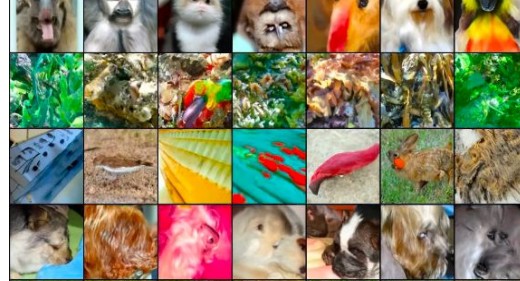

(a) Pretraining with class-balanced mini-batch sampling  (b) Pretraining with class-imbalanced mini-batch sampling

Figure 2: Visualization of prototypes learned by an MSN model pretrained on ImageNet-1K with either class-balanced or class-imbalanced mini-batch distributions. We use RCDM (Bordes et al., 2022b) to enable visualization of the prototypes (details in Appendix H). Each row corresponds to samples generated by conditioning on a prototype with various random seeds. Features that remain constant across the row depict information contained in the prototypes, whereas features that vary depict information that is not contained. a) When pretraining with class-balanced mini-batches, the emergent features tend to be associated with high level concepts, such as specific ImageNet classes. b) By contrast, when pretraining with class-imbalanced mini-batches, the learned features tend to be associated with low-level concepts, such as shape, pose, or texture.

## 5 PRIOR MATCHING FOR SIAMESE NETWORKS

Section 4 validates that in settings where the samples within a mini-batch do not follow a uniform class distribution, joint-embedding methods with volume maximization penalties encode less class-oriented features, and perform worse on downstream semantic classification tasks. In this section, we demonstrate that using alternative (long-tailed) feature priors can lead to representations of a higher semantic level when pretraining on real-world class-imbalanced data.

**Siamese Networks with Arbitrary Prior.** In Assran et al. (2022), the MSN prior is explicitly set to the uniform distribution. As discussed in [Section 3, equation 4], the KL penalty in MSN explicitly encourages learning representations that enable uniform clustering of the data. However, when pretraining with class-imbalanced data, the semantic concepts of interest no longer satisfy the assumptions of the uniformity prior. In particular, natural observations "in the wild" tend to follow long-tailed (often power-law) distributions (Newman, 2005; Mahajan et al., 2018; Van Horn et al., 2018). Based on this observation, we introduce Prior Matching for Siamese Networks (PMSN), which extends MSN to support the use of arbitrary feature priors. Specifically, we modify MSN by replacing the negative entropy term in equation 6 with the KL-divergence to a user-specified distribution.[2] For instance, we can instantiate PMSN as

$$\frac{1}{N} \sum_{i=1}^{N} H(\boldsymbol{p}_i^+, \boldsymbol{p}_i) + \lambda D_{\mathrm{KL}}(\overline{\boldsymbol{p}} \, \| \, \boldsymbol{p}_{\mathrm{PL}(\tau)}), \tag{9}$$

where $\boldsymbol{p}_{\mathrm{PL}(\tau)}$ is a power-law distribution with exponent $\tau > 0$.

### 5.1 TOY SETTING

We claim that the uniform prior in joint-embedding methods significantly impacts the features captured in their learned representations. To illustrate this point to the research community using well-known data, we construct a toy setting to study how changes in the prior affect the suppression and selection of features by the encoder. We take grayscaled CIFAR10 (Krizhevsky et al., 2009) images and overlay one of ten MNIST digits (LeCun & Cortes, 2010) in the top left corner, such that the overall distribution of MNIST digits in the dataset follows a power-law distribution with exponent $0.5$.[3] Next we perform self-supervised pretraining on this dataset using PMSN. We compare pretraining using a uniform feature prior, to pretraining using a power-law prior with exponent equal to $0.5$, which corresponds to the true power-law distribution of MNIST digits in the toy dataset. The first column in Figure 3 shows reference images from this toy dataset; the images in subsequent column visualize the corresponding nearest neighbours in the embedding space of the pretrained models. When pretraining using a power-law prior, the MNIST digit is encoded by the model, and the nearest neighbours all have the same digit class (Figure 3b). However, when pretraining with a uniform prior, MNIST digit information is discarded by the model, and therefore the nearest neighbours have different digit classes (Figure 3c). In particular, since the MNIST digit "feature" does not admit a uniform clustering of the data, it is discarded when pretraining with the conventional volume maximization penalty. This toy experiment provides insight into how semantic features in class-imbalanced datasets can be suppressed by the encoder and how alternative (non-uniform) priors can be used to recover such features. We further show that PMSN with mixture of priors can capture features with different distributions on this task in Appendix I.

### 5.2 NATURAL CLASS-IMBALANCED SETTING

In this section, we examine the downstream task performance obtained by pretraining PMSN in more realistic settings. We examine pretraining with both the IN1K, which is relatively class-balanced, and with the iNaturalist18 dataset (Van Horn et al., 2018), which is relatively class-imbalanced. The iNat18 dataset contains approximately 430K images from over 8142 different species of plants and animals; since some species are more abundant and easier to photograph than others, the class distribution of images in this dataset naturally follows a long-tailed distribution. After pretraining with either iNat18 or IN1k, we examine downstream task performance on the same set of tasks used in Section 4. Additional details about the experimental setup can be found in Appendix E.

---

[2] The negative entropy in equation 6 is simply the KL-divergence to the uniform distribution plus a constant.
[3] Note the image distribution is still uniform over the CIFAR10 classes.

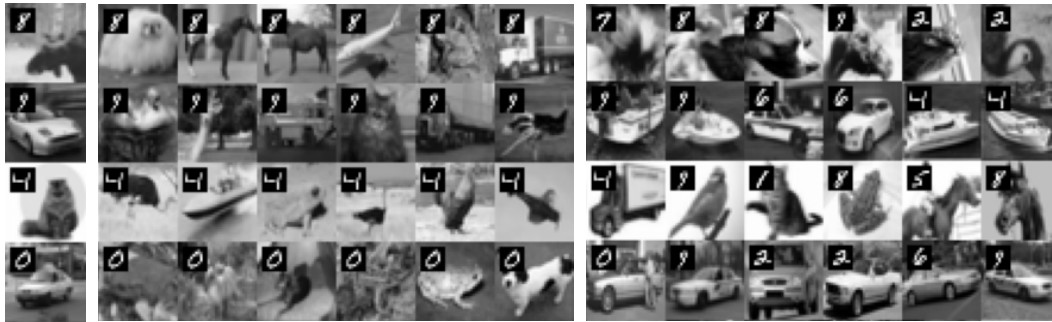

(a) Ref.  (b) Nearest Neighbours when pretraining with power-law prior
**(Matching MNIST digit distribution)**

(c) Nearest Neighbours when pretraining with uniform prior
**(Not matching MNIST digit distribution)**

Figure 3: Each row visualizes the nearest neighbours of the references images (first column), in the embedding space of an PMSN model pretrained on grayscaled images with an MNIST digit in the top left corner. The distribution of MNIST digits in the dataset is constructed to follow a long-tailed power-law distribution. b) When pretraining using a power-law prior, the MNIST digit is encoded by the model, and the nearest neighbours all have the same digit class. c) When pretraining with a uniform prior, the MNIST digit information is discarded by the model, and therefore the nearest neighbours have different digit classes.

**Results with true class distribution.** Table 2 compares iNat18 pretraining with PMSN using a uniform distribution or the *true* class distribution of iNat18. In particular, if we have $D$ images in the dataset and $D_k$ images in the $k^{\text{th}}$ class category, then we define the class-prior, $\boldsymbol{p}_{\text{class}}$, such that $[\boldsymbol{p}_{\text{class}}]_k = D_k/D$. Of course this information is not usually assumed to be known in self-supervised pretraining; we consider it here to illustrate the effect of the uniform prior on representation learning. As shown in Table 2, simply replacing the uniform prior with the long-tailed class-prior improves performance on *all* downstream tasks: object classification with CIFAR100 (+0.9%), iNat18 (+3.0%), Places205 (+1.4%), object counting with Clevr/Count (+2.1%), and depth prediction with Clevr/Dist (+2.3%) and KITTI (+2.7%). These results clearly support the intuition that moving away from the uniformity prior when pretraining with class-imbalanced data can improve the quality of the representations that are learned. However, note that we have used the true class distribution in these experiments. While this distribution may be estimated using weak supervision sources often available for internet-scale data (e.g., image captions or hashtags), we would prefer general methods that do not require precise a priori knowledge of the class distribution.

Table 2: **True Class Prior:** Comparing the uniform prior with the class prior computed from the true class distribution leads to significant gains on downstream tasks.

| Prior | iNat18 | CIFAR100 | Place205 | Clevr/Count | Clevr/Dist | KITTI |
|---|---|---|---|---|---|---|
| **Pretrained on iNaturalist18 (ViT-S/16)** | | | | | | |
| uniform | 29.1 | 59.4 | 36.9 | 69.4 | 56.8 | 68.2 |
| class-prior | **32.1** | **60.3** | **38.3** | **71.5** | **59.1** | **70.9** |
| Δ | +3.0 | +0.9 | +1.4 | +2.1 | +2.3 | +2.7 |

**Results with general power-law distributions.** Table 3 evaluates models pretrained using PMSN on the ImageNet-1K and iNat18 datasets when we do not use precise knowledge of the class distribution. In particular, we explore a power-law prior, $\boldsymbol{p}_{\text{PL}(\tau)}$, such that $[\boldsymbol{p}_{\text{PL}(\tau)}]_k \propto 1/k^\tau$ with power-law exponent $\tau = 0.25$. As expected, use of a power-law prior improves downstream task performance when pretraining on the class-imbalanced iNat18 dataset (top half of Table 3), but degrades performance when pretraining on the class-balanced ImageNet-1K dataset (bottom half of Table 3). These results indicate that it is preferable to match the prior distribution in self-supervised algorithms to the distribution of semantic concepts in the pretraining dataset. In the case of ImageNet pretraining, the uniform prior more closely matches the distribution of classes in the dataset, and thus we expect it to achieve strong downstream task performance in that setting.

While these results support the observation that one can still improve SSL pretraining on class-imbalanced datasets without having a priori knowledge of the class distribution, comparing to Table 2

Table 3: **Power-Law Prior**: PMSN with a power-law prior achieves better downstream performance than uniform prior, when the pretraining dataset has a long-tailed classes distribution. Power-law prior hurts performance for class-balance pretraining dataset. Feature Prior should therefore matches the class-distribution.

| Prior | iNat18 | CIFAR100 | Place205 | Clevr/Count | Clevr/Dist | KITTI |
|---|---|---|---|---|---|---|
| **Pretrained on iNaturalist18 (ViT-S/16)** | | | | | | |
| uniform | 29.1 | 59.4 | 36.9 | 69.4 | 56.8 | 68.2 |
| power-law | 30.1 | 60.1 | 37.7 | 71.1 | 58.9 | 68.2 |
| Δ | +1.0 | +0.7 | +0.8 | +1.7 | +2.1 | +0.0 |
| **Pretrained on ImageNet (ViT-S/16)** | | | | | | |
| uniform | 40.5 | 77.6 | 52.1 | 78.4 | 63.2 | 71.1 |
| power-law | 33.3 | 74.8 | 50.0 | 77.4 | 65.3 | 69.1 |
| Δ | -7.2 | -2.8 | -2.1 | -1.0 | 2.1 | -2.0 |

shows that it would be preferable to use the true class distribution when such information is available. In short, we consider the results of this section as a demonstration of the effect of non-uniform feature priors and do not claim to have completely solved the issue of class-imbalanced pretraining. Similar observations hold for other SSL approach such as SwAV (Caron et al., 2020) as shown in App. J.

**Visualizing learned prototype vectors.**    In Figure 4, we use RCDM to visualize the prototypes of a PMSN model pretrained with either power-law or uniform priors on the iNat18 dataset. The features that emerge when pretraining with a power-law prior are more associated with high level concepts such as specific image classes. For example, one can recognize specific types of birds and plants in Figure 4a, which is not the case with the samples generated using the uniform prior prototypes in Figure 4b. These qualitative results further highlight the effect of the overlooked uniform prior in self-supervised learning with class-imbalanced data.

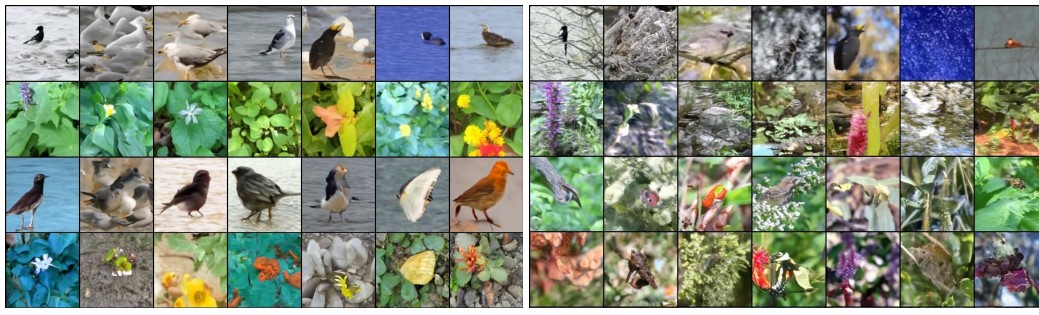

(a) Pretraining on iNat18 with power-law prior
**(Prior matching iNat18 class distribution)**

(b) Pretraining on iNat18 with uniform prior.
**(Prior not aligned with iNat18 class distribution)**

Figure 4: Visualization of prototypes learned by an PMSN model pretrained with either uniform or power-law priors on the iNat18 dataset. We use RCDM (Bordes et al., 2022b) to enable visualization of the prototypes. Each row corresponds to samples generated by conditioning on a prototype with various random seeds. Features that remain constant across the row depicts information contained in the prototypes, whereas features that vary depict information that is not contained. a) Features that emerge when pretraining with a power-law prior on the iNat18 dataset tend to be associated with high level concepts such as specific image classes. b) Features that emerge by pretraining with a uniform prior are largely associated with low-level concepts such as texture.

## 6   Conclusion

In this work, we show that many common self-supervised learning frameworks have a prior to capture features that enable uniform clustering of the data, and as such, require class-balanced datasets to learn class-discriminative features. By reformulating self-supervised criteria to prefer power-law distributed features, one can improve quality of the learned representations on real-world class-imbalanced datasets.

## REPRODUCIBILITY STATEMENT

To facilitate reproducibility, we provide details on our pretraining and evaluation protocol in Appendices D and E. When pretraining using existing methods, we leverage publicly available implementations along with the default hyperparameters; see Appendix D for details. For evaluation, we use the publicly available VISSL codebase (Goyal et al., 2021); specific evaluation configurations are provided in Appendix D.2. The training details for the PMSN experiments are provided in Appendix 5. And finally, the proofs for all propositions in Section 3 are produced in Appendix C.

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

## A  BROADER RELATED WORK

Joint-embedding architectures are an active line of research in self-supervised representation learning (Wu et al., 2018; He et al., 2019; Chen & He, 2020; Grill et al., 2020; Chen et al., 2020c; Caron et al., 2021; Bardes et al., 2021; Zhou et al., 2021). These approaches rely on invariance based pretraining where a neural network encoder is trained to output similar embeddings for two or more views of the same image. To avoid pathological solution, joint-embedding approaches use explicit regularization (Chen et al., 2020b; Caron et al., 2021; Bardes et al., 2021; Assran et al., 2021) or architectural constraints (Grill et al., 2020; Chen & He, 2020).

Explicit regularization usually maximizes the volume of space occupied by the representations. Regularization can be implemented using various strategies such as contrasting negative samples (Bromley et al., 1993; He et al., 2019; Chen et al., 2020b), variance-covariance regularization (Bardes et al., 2021; Zbontar et al., 2021), or by maximizing the entropy of the representations (Asano et al., 2019; Caron et al., 2020; 2021; Assran et al., 2021; 2022). To boost performance of contrastive methods when pre-training with long-tailed data, Zhou et al. (2022) propose to increase the strengths of data-augmentations applied to tail samples. Alternative collapse-prevention approaches based on architectural constraints leverage architectural design to avoid collapse such as stopping the gradient flow in one of the Siamese Network branches (Chen et al., 2020b), using a momentum encoder to compute the network targets (Grill et al., 2020), or using an asymmetric prediction head (Grill et al., 2020; Chen et al., 2020b; Baevski et al., 2022). Recent theoretical work (Tian et al., 2021b) explores why certain joint-embedding methods with architectural constraint avoid representation collapse without explicit use of a volume maximization penalty; the implicit collapse prevention mechanisms here are not mutually exclusive. Recent empirical work (Bordes et al., 2022a) studies the invariance properties of these pretrained representations. The work of Mitrovic et al. (2020) has drawn connections between invariance and causality when the data-augmentations used during pretraining manipulate specific factors of variation in the data-generating distribution. Other works have studied the computational efficiency of joint-embedding methods, and demonstrated how small amounts of supervision can be used to accelerate convergence (Assran et al., 2020).

While joint-embedding architectures are usually leveraged to learn a global image representations, some works explore these use of these architecture for learning local and dense representations (Chen et al., 2022; Gidaris et al., 2020). More recently, LeCun (2022) proposes an architecture based on joint-embedding approaches to learn generic world model, capturing both dense local features as well as global image features.

Orthogonal to the contributions of invariance-based pretraining, another line of work attempts to learn representations by artificially masking parts of the input and training a network to reconstruct the hidden content (Vincent et al., 2010). Auto-regressive models, and denoising auto-encoders, in particular, predict clean visual inputs from noisy views (Chen et al., 2020a; Vincent et al., 2010; He et al., 2021; Bao et al., 2021; Baevski et al., 2022). A mask-noise is usually used to perturb the images and those approaches predict the masked inputs either at the pixel level (Dosovitskiy et al., 2020; He et al., 2021; Xie et al., 2019) or at a token-level, using a pixel (often patch-level) tokenizer (Bao et al., 2021; Wei et al., 2021). While these works demonstrate impressive scalability, they usually learn features at a low-level of semantic abstraction compared to joint-embedding approaches (Assran et al., 2022).

More recently, a set of approaches attempt to combine both joint-embedding and reconstruction based approaches Zhou et al. (2021); El-Nouby et al. (2021), wherein they combine an invariance pretraining loss with a patch-level reconstruction loss.

## B  RELATION TO THE INFOMAX PRINCIPLE

A longstanding conviction in unsupervised representation learning is that the resulting representations should be both maximally informative about the inputs, while also satisfying certain simplicity constraints (Linsker, 1988; Goodfellow et al., 2016). The former objective is often referred to as the information-maximization principle (InfoMax), while the latter is sometimes referred to as the parsimony principle (Ma et al., 2022), which is crucial to the problem formulation. Indeed, Bridle et al. (1991), one of the first works to empirically explore unsupervised representation learning via information-maximization, found that, in the absence of additional constraints, the resulting

InfoMax representations may not be very useful. The more recent analysis of Tschannen et al. (2019) also argues that simplicity constraints are essential to the success of modern representation learning methods built on the InfoMax principle. Historically, simplicity constraints were enforced by encouraging the learned representations to be sparse, low-dimensional, or disentangled, i.e., the individual dimensions of the representation vector should be statistically independent.

Modern state-of-the-art approaches for unsupervised representation learning still frequently employ an information-maximization formulation (Hjelm et al., 2018; Bachman et al., 2019; Krause et al., 2010; Hu et al., 2017; Oord et al., 2018), but with the simplicity constraints manifested in self-supervised loss terms. One example is the widespread view-invariance penalty (Misra & van der Maaten, 2020), often coupled with with independence (Zbontar et al., 2021; Bardes et al., 2021) or low-dimensionality constraints, e.g., by projecting representations on the unit hypersphere (Chen et al., 2020b; He et al., 2019; Grill et al., 2020).

To better understand these methods, recall that the mutual information, $I(\cdot, \cdot)$, between a latent vector $Z$ and data $X$ can be written as

$$I(Z, X) = H[Z] - H[Z|X],$$

where $H[Z]$ is the marginal entropy of $Z$, and $H[Z|X]$ is the expected entropy of the posterior distribution. In learning a representation of the data, $Z$, that maximizes the mutual information $I(Z, X)$, we thereby seek to maximize the marginal entropy of $Z$, i.e., search for uniformly distributed feature embeddings.

However, as we show in this work, perhaps the features that we wish our representations to capture are not necessarily those with the highest marginal entropy. It is often the case that the semantic concepts we wish to capture actually follow a long-tailed distribution in the wild . Such desirable features would be penalized under existing information-maximization frameworks. Thus, in the absence of finer grained notions of information, perhaps it is necessary to reconsider the longstanding conviction of seeking representations that maximize information content.

## C  THEORETICAL GUARANTEES

### C.1  PROOF OF PROPOSITION 1

We will demonstrate that

$$\min_{\mu_1,\ldots,\mu_k} \sum_{n=1}^{N} \min_{c=1,\ldots,K} \|\boldsymbol{x}_n - \mu_c\|_2^2 = \min_{\boldsymbol{P} \in \{0,1\}^{N \times K}: \boldsymbol{P}\mathbf{1}_K = \mathbf{1}_N} \sum_{k=1}^{K} \sum_{n,n'=1}^{N} \frac{\boldsymbol{P}_{n,k} \boldsymbol{P}_{n',k}}{\mathbf{1}_N^T \boldsymbol{P}_{.,k}} \|\boldsymbol{x}_n - \boldsymbol{x}_{n'}\|_2^2,$$

(10)

where the left-hand side minimizes over the values of the $K$ centroids and the inner minimization identifies the cluster membership of sample $x_n$. To do so, one should first notice that almost surely, for any input $\boldsymbol{x}_n$, the $\min_{c=1,\ldots,K} \|\boldsymbol{x}_n - \mu_c\|_2^2$ is attained for a single centroid. In fact, for any continuous distribution on the data (and/or centroids) the probability to sample them so that a sample $\boldsymbol{x}$ lies exactly equidistant from two (or more) centroids is 0 since this is a space of dimension $D - 1$, which has measure 0. Hence, we first obtain

$$\min_{\mu_1,\ldots,\mu_k} \sum_{n=1}^{N} \min_{c=1,\ldots,K} \|\boldsymbol{x}_n - \mu_c\|_2^2 = \min_{\mu_1,\ldots,\mu_k} \sum_{n=1}^{N} \min_{\boldsymbol{p} \in \{0,1\}^K: \boldsymbol{p}^T \mathbf{1}_K = 1} \sum_{k=1}^{K} \boldsymbol{p}_k \|\boldsymbol{x}_n - \mu_k\|_2^2,$$

and because each sub-problem is independent, we can pull them out of the sum to obtain

$$\min_{\mu_1,\ldots,\mu_k} \sum_{n=1}^{N} \min_{c=1,\ldots,K} \|\boldsymbol{x}_n - \mu_c\|_2^2 = \min_{\mu_1,\ldots,\mu_k} \min_{\boldsymbol{P} \in \{0,1\}^{N \times K}: \boldsymbol{P}\mathbf{1}_K} \sum_{n=1}^{N} \sum_{k=1}^{K} \boldsymbol{P}_{n,k} \|\boldsymbol{x}_n - \mu_k\|_2^2,$$

now we can switch the minimization order to obtain

$$\min_{\boldsymbol{P} \in \{0,1\}^{N \times K}: \boldsymbol{P}\mathbf{1}_K} \min_{\mu_1,\ldots,\mu_k} \sum_{n=1}^{N} \sum_{k=1}^{K} \boldsymbol{P}_{n,k} \|\boldsymbol{x}_n - \mu_k\|_2^2 = \min_{\boldsymbol{P} \in \{0,1\}^{N \times K}: \boldsymbol{P}\mathbf{1}_K} \sum_{n=1}^{N} \sum_{k=1}^{K} \boldsymbol{P}_{n,k} \|\boldsymbol{x}_n - \mu_k^*\|_2^2,$$

with $\mu_k^*$ the mean of the samples within cluster $k$, i.e. $\frac{1}{(P^T \mathbf{1}_N)_k} \sum_{n=1}^N P_{n,k} x_n$. The only thing left to show is that the sum of distances to the cluster means is equivalent to the pairwise distances between all the points within each cluster with an appropriate normalization, which is a standard result; see, e.g., Zha et al. (2001).

## C.2 Proof of Proposition 2

**Notation.** Before establishing the theoretical connections between common joint-embedding methods for self-supervised learning and the K-means method, we first define some notation to facilitate the discussion. Given a "stand-alone" dataset, $X' \in \mathbb{R}^{N' \times 3HW}$, we construct the pretraining data $X \in \mathbb{R}^{N \times 3HW}$ by repeatedly perturbing the elements of the stand-alone dataset. Specifically,

$$X \triangleq [\text{View}_1(X')^T, \ldots, \text{View}_V(X')^T]^T,$$

where $\text{View}_i(.)$ is a sample-wise image transformation, e.g., random crop, color jitter, patch masking. We also define the ground-truth similarity matrix, $G \in \{0,1\}^{N \times 3HW}$, given by

$$G_{i,j} = \begin{cases} (x_i \sim x_j), & i \neq j \\ 0, & i = j \end{cases},$$

where the $\sim$ operator returns 1 if its two arguments are positively related (i.e., correspond to different views of the same sample). We also define the matrix of *embeddings* obtained from a model $f_\theta(\cdot)$ as

$$Z \triangleq [z_1, \ldots, z_N]^T \in \mathbb{R}^{N \times D} \quad \text{with} \quad z_n \triangleq f_\theta(x_n). \tag{11}$$

Now we are ready to present our first equivalence relations for the VICReg and SimCLR methods. The VICReg loss (Bardes et al., 2021), which is a function of $X$ and $G$, can be defined as

$$\mathcal{L} = \alpha \sum_{k=1}^K \max\left(0, 1 - \sqrt{\text{Cov}(Z)_{k,k}}\right) + \beta \sum_{k=1}^K \sum_{\substack{\ell=1 \\ \ell \neq k}}^K \text{Cov}(Z)_{k,\ell}^2 + \frac{\gamma}{N} \sum_{i=1}^N \sum_{j=1}^N (G)_{i,j} \|z_i - z_j\|_2^2,$$

but this desired result relies on the simplification employed in equation 3. To prove our statement, we first remind the reader of a common result that we will heavily rely on: the decomposition of the covariance matrix into within- and between-cluster covariance matrices. Let us assume for simplicity that $Z$ is already centered. We can now decompose the covariance into

$$\text{Cov}(Z) = \frac{1}{N} Z^T H Z = \frac{1}{N}\left(Z^T G Z + Z^T(I - G)Z\right),$$

where two terms are now the *between cluster* and *within cluster* covariances, with $H$ the centering matrix defined by $I - \mathbf{1}_N \mathbf{1}_N^T / N$. Let's first consider the LHS of the VICReg loss to simplify it into

$$\left\|\frac{1}{N} Z^T Z - I\right\|_F^2 = \left\|\frac{1}{N} Z^T G Z + \frac{1}{N} Z^T(I - G)Z - I\right\|_F^2$$

$$= \left\|\frac{1}{N} Z^T G Z - I\right\|_F^2 + \left\|\frac{1}{N} Z^T(I - G)Z\right\|_F^2$$

$$+ \frac{2}{N} Tr\left(Z^T(I - G)Z(\frac{1}{N} Z^T G Z - I)\right)$$

$$= \left\|\frac{1}{N} Z^T G Z - I\right\|_F^2 + \left\|\frac{1}{N} Z^T(I - G)Z\right\|_F^2 - \frac{2}{N} Tr\left(Z^T(I - G)Z\right)$$

$$+ \frac{2}{N^2} Tr\left(Z^T(I - G)Z Z^T G Z\right)$$

$$= \left\|\frac{1}{N} Z^T G Z - I\right\|_F^2 + \sum_i \lambda_i\left(\frac{1}{N} Z^T(I - G)Z\right)\left[\lambda_i\left(\frac{1}{N} Z^T(I - G)Z\right) - 2\right]$$

$$+ \frac{2}{N^2} Tr\left(Z^T(I - G)Z Z^T G Z\right),$$

where $\lambda_i(\boldsymbol{M})$ returns the $i^{\text{th}}$ eigenvalue of its matrix argument $\boldsymbol{M}$; the last equality is obtained by noticing that $\|\boldsymbol{M}\|_F^2 = \sum_i \lambda_i(\boldsymbol{M})^2$ and that for a symmetric positive semidefinite matrix, $Tr(\boldsymbol{M}) = \sum_i \lambda_i(\boldsymbol{M})$; we thus obtain the following upper and lower bounds

$$\left\|\frac{1}{N}\boldsymbol{Z}^T\boldsymbol{Z} - \boldsymbol{I}\right\|_F^2 \leq \left\|\frac{1}{N}\boldsymbol{Z}^T\boldsymbol{G}\boldsymbol{Z} - \boldsymbol{I}\right\|_F^2 + \sum_i \lambda_i\left(\frac{1}{N}\boldsymbol{Z}^T(\boldsymbol{I}-\boldsymbol{G})\boldsymbol{Z}\right)$$
$$\times \left[\lambda_i\left(\frac{1}{N}\boldsymbol{Z}^T(\boldsymbol{I}-\boldsymbol{G})\boldsymbol{Z}\right) - 2 + \lambda_i\left(\frac{1}{N}\boldsymbol{Z}^T\boldsymbol{G}\boldsymbol{Z}\right)\right]$$

$$\left\|\frac{1}{N}\boldsymbol{Z}^T\boldsymbol{Z} - \boldsymbol{I}\right\|_F^2 \geq \left\|\frac{1}{N}\boldsymbol{Z}^T\boldsymbol{G}\boldsymbol{Z} - \boldsymbol{I}\right\|_F^2 + \sum_i \lambda_i\left(\frac{1}{N}\boldsymbol{Z}^T(\boldsymbol{I}-\boldsymbol{G})\boldsymbol{Z}\right)$$
$$\times \left[\lambda_i\left(\frac{1}{N}\boldsymbol{Z}^T(\boldsymbol{I}-\boldsymbol{G})\boldsymbol{Z}\right) - 2 + \lambda_{K+1-i}\left(\frac{1}{N}\boldsymbol{Z}^T\boldsymbol{G}\boldsymbol{Z}\right)\right]$$

$$\left\|\frac{1}{N}\boldsymbol{Z}^T\boldsymbol{Z} - \boldsymbol{I}\right\|_F^2 \geq \left\|\frac{1}{N}\boldsymbol{Z}^T\boldsymbol{G}\boldsymbol{Z} - \boldsymbol{I}\right\|_F^2 + \sum_i \left[\lambda_i\left(\frac{1}{N}\boldsymbol{Z}^T(\boldsymbol{I}-\boldsymbol{G})\boldsymbol{Z}\right) - 1\right]^2 - K,$$

from which is becomes clear that to minimize the variance+covariance terms, one must either maximize the intra-cluster variance, or the inter-cluster variance, or both. However, the intra-cluster variance is exactly the invariance term since it can be expressed as

$$\sum_{j=1}^{N}(\boldsymbol{G})_{i,j}\|\boldsymbol{z}_i - \boldsymbol{z}_j\|_2^2 = 2Tr(\boldsymbol{Z}^T(\boldsymbol{I}-\boldsymbol{G})\boldsymbol{Z}),$$

and thus the only possible solution to minimize the invariance term while minimizing the variance+covariance is to minimize $\left\|\frac{1}{N}\boldsymbol{Z}^T\boldsymbol{G}\boldsymbol{Z} - \boldsymbol{I}\right\|_F^2$ and thus we recover that VICReg's loss corresponds to the K-means loss plus a regularizer $\|\mu\mu^T - \boldsymbol{I}\|_F^2$ as in

$$\frac{\gamma}{N}\sum_{k=1}^{K}\sum_{\boldsymbol{x}\in\mathbb{X}_k}\|\boldsymbol{x} - \mu_k\|_2^2 + \alpha\|\boldsymbol{M}^T\boldsymbol{M} - \boldsymbol{I}\|_F^2,$$

with $\boldsymbol{M} \triangleq [\mu_1, \ldots, \mu_K]$, and the number of centroids $K$ is given by $\min(\dim(\boldsymbol{z}), rank(\boldsymbol{G}+\boldsymbol{I}))$ and the centroids are given by $\mu_k = \frac{1}{\langle\boldsymbol{P}_{\cdot,k},\boldsymbol{1}\rangle}\sum_{n=1}^{N}\boldsymbol{P}_{n,k}\boldsymbol{x}_n$ with $\boldsymbol{G} = \boldsymbol{P}^T\boldsymbol{D}\boldsymbol{P}$, and finally $\mathbb{X}_k = \{\boldsymbol{x}_n \in \mathbb{X} : \boldsymbol{P}_{n,k} > 0\}$

### C.3 PROOF OF PROPOSITION 3

The proof is relatively straightforward and will follow the same principle as the proof showing how GMM recovers K-means. The only difference is that we will take the zero-noise limit of the MSN loss (equation 6) instead of GMM loss (equation 5), and we will see that we recover a constrained version of K-means with extra constraints on the cluster distribution. The MSN loss is defined for positive pairs $\boldsymbol{x}_n, \boldsymbol{x}_n^+$ and estimates the corresponding cluster posteriors $\boldsymbol{p}_n, \boldsymbol{p}_n^+$ via softmax$(\boldsymbol{M}^T\boldsymbol{x}_n/\sigma)$ and softmax$(\boldsymbol{M}^T\boldsymbol{x}_n^+/\sigma^+)$ respectively with $\boldsymbol{M} \triangleq [\mu_1, \ldots, \mu_K]$ as we employed in VICReg, and the two temperature parameters are commonly $\sigma^+ \gg \sigma > 0$; this asymmetry is know as sharpening. Without loss of generality we consider the $\sigma$MSN loss, i.e., the MSN loss re-scaled by $\sigma$, this does not alter the training dynamics as the learning rate can be adapted accordingly, but will simplify our derivations below; we thus also replace $\lambda$ with $\lambda/\sigma$. In this setting, we see that softmax$(\boldsymbol{M}^T\boldsymbol{x}_n^+/\sigma^+)_k = \delta(k - k(n))$ where we hereafter denote $k(n) = \arg\min_c \|\mu_c - \boldsymbol{x}_n^+\|_2$ as the cluster assignment of the positive view of $\boldsymbol{x}_n$. With those notations, we can finally derive

$$\lim_{\sigma\mapsto 0}\sigma MSN = \lim_{\sigma\mapsto 0}\left[\frac{\sigma}{N}\sum_{n=1}^{N}H(\boldsymbol{p}_n^+, \boldsymbol{p}_n) + \lambda D_{\text{KL}}(\overline{\boldsymbol{p}} \,\|\, \boldsymbol{p}_{\text{prior}})\right]$$

$$= \lim_{\sigma\mapsto 0}\left[-\frac{\sigma}{N}\sum_{n=1}^{N}\sum_{k=1}^{K}\delta(k - k(n))\log\left(\text{softmax}(\boldsymbol{W}^T\boldsymbol{x}_n/\sigma)_k\right)\right] + \lambda\sum_{k=1}^{K}\frac{N_k}{N}\log\left(\frac{N_k/N}{(\boldsymbol{p}_{\text{prior}})_k}\right)$$

where we could push the limit inside $D_{\text{KL}}$ since we assume that no cluster is empty and thus the KL function is continuous as $\sigma \mapsto 0$, and that $\lim_{\sigma \mapsto 0} \overline{\boldsymbol{p}} = (\frac{N_1}{N}, \ldots, \frac{N_K}{N})$. The value of $N_k$ is the number of samples that are assigned to cluster $k$. This assumption, which was only used for simplicity, can be removed easily by noticing that $\lim_{u \mapsto 0} u \log(u) = 0$. Now considering the left term of the loss we obtain a direct simplification of the log-softmax as follows

$$\lim_{\sigma \mapsto 0} \left[ -\frac{\sigma}{N} \sum_{n=1}^{N} \sum_{k=1}^{K} \delta(k - k(n)) \log \left( \text{softmax}(\boldsymbol{W}^T \boldsymbol{x}_n / \sigma)_k \right) \right]$$

$$= \frac{1}{N} \sum_{n=1}^{N} \underbrace{\|\boldsymbol{x}_n - \mu_{k(n)}\|_2^2 - \min_{k=1,\ldots,K} \|\boldsymbol{x}_n - \mu_k\|_2^2}_{=0 \iff \arg\min_c \|\boldsymbol{x}_n - \mu_c\|_2 = \arg\min_c \|\boldsymbol{x}_n^+ - \mu_c\|_2} ,$$

which is minimized whenever all the samples $\boldsymbol{x}_n$ that have their positive views $\boldsymbol{x}_n^+$ associated to the same centroid belong to the same cluster; thus the overall MSN loss can be written as

$$\sum_{k=1}^{N} \sum_{\boldsymbol{x} \in \mathbb{X}_k} \|\boldsymbol{x} - \mu_k\|_2^2 + \lambda \sum_{k=1}^{K} \frac{N_k}{N} \log \left( \frac{N_k/N}{(\boldsymbol{p}_{\text{prior}})_k} \right),$$

where $\mathbb{X}_k = \{\boldsymbol{x} \in \mathbb{X} : \arg\min_{c=1,\ldots,K} \|\boldsymbol{x}^+ - \mu_c\|_2 = k\}$.

### C.4    PROOF OF PROPOSITION 4

The proof for SwAV mainly relies on the same development as done for MSN. That is, we saw that with enough sharpening, the cross-entropy between $\boldsymbol{p}_n^+$ and $\boldsymbol{p}_n$ falls back to the K-means like term with an extra margin that is minimized only when the two views are in the same cluster. The difference arises in that SwAV explicitly adds linear constraints on the cluster-membership matrix whereas MSN was employing a KL-divergence between the average posterior and the prior. The connection between the linear constraint and the SwAV Sinkhorn-Knopp procedure has been made precisely in Wang et al. (2010), where it was shown that the latter solved a relaxed optimization problem for which the cluster-membership is no longer constrained to be $0$ or $1$.

## D    PRETRAINING AND EVALUATION DETAILS FOR SECTION 4

### D.1    PRETRAINING PROTOCOL

**SimCLR.**  We use the VISSL (Goyal et al., 2021) code base to pretrain a ResNet-50 with Sim-CLR (Chen et al., 2020b), with a batch size of 4096 for 300 epochs. Our pretraining follow the standard hyperparameters defined in Chen et al. (2020b). The learning rate follows the default cosine schedule with a 10 epoch warmup. We use a temperature of 0.1 for the contrastive loss and LARS (You et al., 2017) as an optimizer. We modify the sampler to force $K$ different classes inside each mini-batch where $K$ is set to 8 for the class-imbalanced sampling experiments and 960 for the class-balanced sampling experiments.

**VICReg.**  We pretrain a ResNet-50 with VICReg (Bardes et al., 2021) using the LARS optimizer with a batch size of 1024 for 300 epochs using the official code base, which is publicly available: https://github.com/facebookresearch/vicreg. Our pretraining follow the standard hyperparameters defined in Bardes et al. (2021). The learning rate follows the default cosine schedule with a 10 epoch warmup. We set the dimensions of the expander MLP to the default $8192 - 8192 - 8192$. Weight decay is set to $10^{-6}$, the variance and invariance coefficients are set to 25.0, and the covariance coefficient is set to 1.0. We modify the sampler to force $K$ different classes inside each mini-batch where $K$ is set to 2 for the class-imbalanced sampling experiments and 960 for the class-balanced sampling experiments.

**MSN.**  We pretrain a ViT-B/16 with MSN (Assran et al., 2022) using the AdamW optimizer with a batch size of 1024 for 300 epochs and 1024 prototypes using the official code base, which is publicly available: https://github.com/facebookresearch/msn. Our pretraining follow the

standard hyperparameters defined in Assran et al. (2022). The learning rate follows the default cosine schedule with a 15 epoch warmup. Weight decay is linearly increased from $0.04$ to $0.4$. Gradient clipping is set to $3.0$, the entropy coefficient is set to $1.0$, the temperature is set to $0.1$, the sharpening exponent is set to $0.25$, and the masking ratio is set to $0.5$. The number of random-mask views in each iteration is set to 1, and the number of focal-mask views in each iteration is set to 10. We modify the sampler to force $K$ different classes inside each mini-batch where $K$ is set to 2 for the class-imbalanced sampling experiments and 960 for the class-balanced sampling experiments.

**MAE.** We pretrain a ViT-L/16 with MAE (He et al., 2021) using the AdamW optimizer with a batch size of 1024 for 800 epochs using the official code base, which is publicly available: `https://github.com/facebookresearch/mae`. Our pretraining follow the standard hyperparameters defined in He et al. (2021). The learning rate follows the default cosine schedule with a 40 epoch warmup. Weight decay is set to $0.05$, and the masking ratio is set to $0.75$. We modify the sampler to force $K$ different classes inside each mini-batch where $K$ is set to 2 for the class-imbalanced sampling experiments and 960 for the class-balanced sampling experiments.

**data2vec.** We pretrain a ViT-B/16 with data2vec (He et al., 2021) using the AdamW optimizer with a batch size of 2048 for 800 epochs using the official code base, which is publicly available: `http://github.com/facebookresearch/data2vec_vision/tree/main/beit`. Our pretraining follow the standard hyperparameters defined in Baevski et al. (2022). Specifically, the learning rate follows the default cosine schedule with a 10 epoch warmup. Path drop is set to $0.25$, gradient clipping is set to $3.0$, weight decay is set to $0.05$, and the target layers are set to $[6, 7, 8, 9, 10, 11]$. We modify the sampler to force $K$ different classes inside each mini-batch where $K$ is set to 2 for the class-imbalanced sampling experiments and 960 for the class-balanced sampling experiments.

## D.2 EVALUATION PROTOCOL

For linear evaluation, we use the default linear evaluation configurations of VISSL (Goyal et al., 2021) to evaluate our models on the following datasets: ImageNet (Russakovsky et al., 2015), iNaturalist18 (Van Horn et al., 2018), CIFAR100 (Krizhevsky et al., 2009), Clevr/Count (Johnson et al., 2017), Clevr/Dist (Johnson et al., 2017), KITTI/Dist (Geiger et al., 2013) and Places205 (Zhou et al., 2014).

For pretrained models based on Vision Transformers (Dosovitskiy et al., 2020), we report the best linear classifier number among the following representations:

- the concatenation of the last 4 layers of the class token, (Caron et al., 2021)
- the representation of the last layer of the class token.

For pretrained models based on ResNet50 architectures (He et al., 2016), we follow the evaluation protocol of SEER (Goyal et al., 2022) and report the best linear classifier number among the following representations:

- the final representation layer (of dimension 2048 for a ResNet50),
- an adaptive average pooling of the last feature map, concatenated to get 8192 dimensions.

We also follow the default VISSL (Goyal et al., 2021) configuration and attach 2 linear heads per chosen representation, one composed of a single linear layer, and one with an added batch normalization (Ioffe & Szegedy, 2015) before the linear layer.

## E   PRETRAINING AND EVALUATION DETAILS FOR SECTION 5

### E.1   PRETRAINING PROTOCOL

#### E.1.1   TOY SETTING

We pretrain a ViT-Tiny/4 with MSN (Assran et al., 2022) using the AdamW optimizer with a batch size of 1024 for 300 epochs and 10 prototypes using the official code base, which is publicly

available: `https://github.com/facebookresearch/msn`. The learning rate follows the default cosine schedule with a 15 epoch warmup. Weight decay is linearly increased from 0.04 to 0.4. Gradient clipping is set to 0.0, the entropy coefficient is set to 100.0, the temperature is set to 0.1, the sharpening exponent is set to 0.25, and the masking ratio is set to 0.05. The number of random-mask views in each iteration is set to 1, and the number of focal-mask views in each iteration is set to 0. To accommodate the lower-resolution images, we modify the scale of random-resized-crop data augmentation to $(0.5, 1.0)$ and train without Gaussian-Blur.

### E.1.2  Natural class-imbalanced setting

We pretrain a ViT-S/16 with MSN (Assran et al., 2022) using the AdamW optimizer with a batch size of 4096 for 300 epochs and 8142 prototypes using the official code base, which is publicly available: `https://github.com/facebookresearch/msn`. The learning rate follows the default cosine schedule with a 15 epoch warmup. Weight decay is linearly increased from 0.04 to 0.4. Gradient clipping is set to 3.0, the entropy coefficient is set to 5.0, the temperature is set to 0.1, the sharpening exponent is set to 0.25, and the masking ratio is set to 0.15. The number of random-mask views in each iteration is set to 1, and the number of focal-mask views in each iteration is set to 10.

### E.2  Evaluation protocol

We use the exact same evaluation protocol as D.2.

## F  Alternatives to feature priors for section 5: data sampling

As an alternative to changing the feature prior, perhaps one can devise more intelligence sampling strategies to align the class distribution in the sampled mini-batches with the implicit uniform prior in self-supervised algorithms. Here, we investigate the impact on the mini-batch sampling schemes when pretraining on iNat18. We compare (uniform) random sampling to unthresholded inverse square-root frequency sampling (Mikolov et al., 2013), which is commonly applied in the context of (weakly) supervised learning on internet-scale class-imbalanced data. Specifically, the prescription of inverse square-root frequency sampling is to sample a class with probability inversely proportional to the square root of the frequency of the class, and then sample images uniformly within the class. For example, if we have $D$ images in the dataset and $D_k$ images in the $k^{\text{th}}$ class category, then the probability of sampling class $k$ is equal to $\sqrt{D_k}/D$, as opposed to the traditional $D_k/D$ probability under uniform sampling. The effect of this sampling strategy is to mitigate class-imbalances by oversampling underrepresented classes in the dataset. While such an effect is desirable, one limitation to this strategy is that it induces an implicit reduction in the dataset size; for example, we may see the same images from the tail of the distribution very frequently and, given finite training time, may never see some of the images in the head. Another limitation of this strategy is that requires knowing the class label (or a weak class label) for every image in the training set.

Despite these limitations, we explore inverse frequency sampling in Table 4 and observe that it does not improve upon uniform random sampling. We hypothesize the inverse frequency sampling tends to aggressively over-sample classes with very few examples, reducing the effective number of images seen per epoch and thus degrading the quality of the learned representations. This result suggests that changing the feature prior is a more viable solution for pretraining with class-imbalanced data.

Table 4: **Impact of Sampling Scheme**: Changing the mini-batch distribution by using an inverse square root sampling strategy on iNat18 does not improve downstream performance of the learned representations.

| Sampling | iNat18 | CIFAR100 | Place205 | Clevr/Count | Clevr/Dist | KITTI |
|---|---|---|---|---|---|---|
| *Pretrained on iNaturalist18 (ViT-S/16)* | | | | | | |
| Uniform | **29.1** | **59.4** | **36.9** | **69.4** | 56.8 | **68.2** |
| Inverse Square Root Freq | 23.4 | 58.4 | 35.1 | 68.3 | **59.8** | 66.6 |

## G  MARGINAL SAMPLING PROBABILITIES

The class-imbalanced strategy in Section 4 maintains the same marginal probability of sampling individual data points as compared to the class-balanced sampling strategy; i.e., the probability of sampling a particular data point in each iteration is unchanged. To see this, suppose we have a mini-batch of size $B = n \times 960$ for some integer $n > 0$. Under class-balanced sampling, we will thus sample 960 classes in each iteration, and then $n$ images per class. Under class-imbalanced sampling, we will first sample 2 classes in each iteration, and then $n \times 480$ images per class so that the overall batch size is $B$. Now consider the probability of sampling a data point $x$ that comes form a class $C$ in our dataset containing $N \geq 480 \cdot n$ samples. Under class-balanced sampling, the probability of sampling the data point $x$ can be factored as

$$p_{\text{balanced}}(x) = p(x|C)p(C) = \frac{\binom{N-1}{n-1}}{\binom{N}{n}} \frac{\binom{999}{959}}{\binom{1000}{960}} = \frac{n}{N} \frac{960}{1000}.$$

Under class-imbalanced sampling, the probability of sampling $x$ can be factored as

$$p_{\text{imbalanced}}(x) = p(x|C)p(C) = \frac{\binom{N-1}{480 \cdot n-1}}{\binom{N}{480 \cdot n}} \frac{\binom{999}{1}}{\binom{1000}{2}} = \frac{480 \cdot n}{N} \frac{2}{1000},$$

from which it is clear that $p_{\text{balanced}}(x)$ is equal to $p_{\text{imbalanced}}(x)$.

## H  VISUALIZING PROTOTYPES WITH RCDM

We use the RCDM framework (Bordes et al., 2022b) to visualize the representations and prototypes learned with MSN. RCDM trains a conditional generative diffusion model, which maps a noise vector to pixel space using a neural network representation as conditioning. Many works (Ramesh et al., 2022; Saharia et al., 2022) have demonstrated the potential of conditional diffusion model for image generation, however they are also very useful, as highlighted by (Bordes et al., 2022b) to get a better understanding of what is learned by neural networks.

During training, RCDM takes as input a noisy image $\hat{x}_t$ (corrupted with an $\epsilon_t$ noise vector such as $\hat{x}_t = x + \epsilon_t$) and the representation vector $y$ computed by MSN of the image $x$. Then, RCDM is train, with a denoising score matching loss (Vincent, 2011; Ho et al., 2020), to reconstruct the image $x$ that was used to compute the representation vector $y$. More formally, we define a RCDM neural network $g_\eta(\hat{x}_t, y)$ that learns to predict the noise component $\epsilon_t$ of $\hat{x}_t$, ie. by minimizing $\|g_\eta(\hat{x}_t, y) - \epsilon_t\|_2^2$. As demonstrated by (Bordes et al., 2022b), RCDM extract as many information as possible from the representation vector $y$ in order to reconstruct faithfully the image.

The conditioning vector $y$ is computed from $x$ using a pretrained and frozen MSN model. MSN generates as output the probability distribution $p$ that an image $x$ belongs to a given cluster, $p = \text{softmax}(W^t p_\gamma(f_\theta(x)))$ where $W$ is the matrix concatenating all the prototypes (or cluster centroids), $p_\gamma$ the projection head and $f_\theta$ the encoder. To visualize the prototype contained in $W$, we first train a conditional generative diffusion model that take the last linear layer input as conditioning, i.e $y = p_\gamma(f_\theta(x))$. After training, we replace the image embedding with a learned prototype and generate the corresponding output in pixel space by setting $y = W_i$, where $i$ is randomly selected.

To summarize, we gather for every images in the training set their embedding (with dimension size of 256)[4] of a trained MSN model. Then, we use these embedding as conditioning for RCDM which is train to reconstruct the corresponding image associated to a given embedding. When training is complete, we replace the projector's embedding by the prototype learned with MSN (which also have a dimension of 256). By doing so we can visualize which information is associated to each prototype (or cluster) learned with MSN. For every RCDM training, we used the same defaults settings as the ones on https://github.com/facebookresearch/RCDM. We train each network for 200000 iterations.

---

[4]the one that is use to perform the clustering with respect to the prototypes

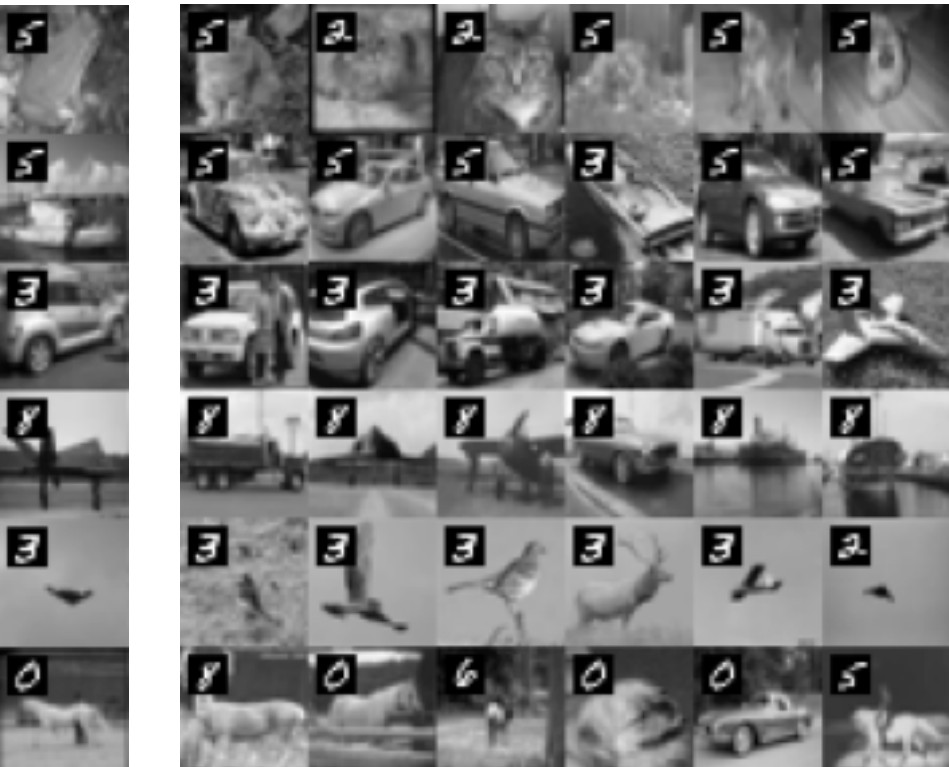

Figure 5: Each row visualizes the nearest neighbours of the references images (first column), in the embedding space of an PMSN model pretrained on grayscaled images with an MNIST digit in the top left corner. The distribution of MNIST digits in the dataset is constructed to follow a long-tailed power-law distribution. Here we consider a PMSN model using two projection heads, one with a power-law prior and one with an uniform prior. Nearest neighbours show both consistent digit and object classes. When pretraining a PMSN with a mixture of priors, the learned representation is able to capture both power-law features (related to the digits) and uniform features (related to the object classes).

## I  MIXTURE OF PRIORS

Section 5.1 illustrated the effect of feature priors in a toy setting. Specifically, we take grayscaled CIFAR10 images and overlay one of ten MNIST digits in the top left corner, such that the overall distribution of MNIST digits in the dataset follows a power-law distribution with exponent $0.5$. When pretraining using MSN with a power-law prior, Figure 3b shows that the digit information is encoded by the model, while the CIFAR10 class information, which is uniformly distributed, is discarded. By contrast, Figure 3c shows that when pretraining using MSN with a uniform prior, the digit information is discarded by the model, whereas the CIFAR10 class information is relatively well captured. An important question to ask is whether it is possible to pretrain with a mixture of priors, so as to capture multiple sets of features following different distributions in the pretraining data.

We follow the experimental setup described in Section 5.1, but pretrain using two different heads. The first head enforces a uniform prior, while the second head enforces a power-law prior. The total loss is simply the sum of the losses from the two heads. The first column in Figure 5 shows reference images from this toy dataset; images in subsequent columns visualize the corresponding nearest neighbours in the embedding space of the pretrained models. Remarkably, using this simple multi-head mixture prior, the model learns to capture both digit information (power-law distributed) and the object class information (uniformly distributed). Indeed, the nearest neighbours tend to exhibit both consistent digit and object classes.

## J   PRIOR MATCHING WITH SwAV

The self-supervised SwAV (Caron et al., 2020) method has an implicit uniform prior through the use of the Sinkhorn-Knopp normalization. With a minor tweak, we can re-parameterize their default Sinkhorn-Knopp normalization to rebalance the mini-batch prior to a power-law instead of a uniform distribution.

We train SwAV for 100 epochs, once with the default implicit uniform prior, and once with a power-law prior with exponent $0.5$. We run these two pretraining runs on both the ImageNet dataset, where the class distribution is relatively balanced, and on the iNaturalist18 datasets, where the class distribution is long-tailed.[5]

Table 6 summarizes these SwAV prior pretraining results on various downstream tasks, and exhibits similar findings as with PMSN: aligning the prior to better match the underlying class distribution increases the quality of the learned representations. SwAV pretrained on ImageNet benefits from a uniform prior, while SwAV pretrained on iNat18 benefits from a power-law prior, boosting in-distribution performance by $1.5\%$ as well as boosting performance on semantic tasks.

Table 5: **Impact of prior on SwAV**: Changing the prior of SwAV by changing the Sinkhorn-Knopp target impacts the representations learned by SwAV. Aligning the prior with the underlying class distribution of the pretraining dataset increases the quality of the learned representations.

| Prior | iNat18 | CIFAR100 | Place205 | Clevr/Count | Clevr/Dist | KITTI |
|---|---|---|---|---|---|---|
| **Pretrained on iNaturalist18 (ResNet50)** | | | | | | |
| uniform | 33.8 | 50.2 | 41.9 | 70.7 | 70.6 | 70.3 |
| power-law | 35.3 | 51.2 | 42.1 | 72.2 | 68.6 | 70.5 |
| Δ | +1.5 | +1.0 | +0.2 | +1.5 | -2.0 | +0.2 |
| **Pretrained on ImageNet (ResNet50)** | | | | | | |
| uniform | 44.4 | 75.6 | 54.5 | 78.4 | 68.6 | 72.5 |
| power-law | 42.6 | 74.2 | 54.6 | 76.6 | 67.9 | 72.3 |
| Δ | -1.8 | -1.4 | +0.1 | -1.8 | -0.7 | -0.2 |

## K   ADDITIONAL RESULTS WITH PMSN

In this section we report additional results with PMSN when pretraining a larger ViT-B/16 encoder with either a power-law or a uniform prior on the class-balanced ImageNet dataset. Results are consistent with Table 3, indicating that the power-law prior can result in less semantic representations when pretraining with class-balanced data.

Table 6: **VIT-B/16 with Power-Law prior.** Power-law prior can result in less semantic representations when pretraining with class-balanced data.

| Prior | iNat18 | CIFAR100 | Place205 | Clevr/Count | Clevr/Dist | KITTI |
|---|---|---|---|---|---|---|
| **Pretrained on ImageNet (ViT-B/16)** | | | | | | |
| uniform | 41.9 | 81.7 | 54.0 | 74.3 | 57.8 | 73.3 |
| power-law | 22.0 | 63.8 | 38.3 | 71.4 | 65.8 | 66.6 |
| Δ | -19.9 | -17.8 | -15.7 | -2.9 | +8.0 | -6.7 |

## L   FULL SET OF RESULTS FOR SECTION 4

In this section, we report the full experimental results for Section 4. Figure 6 shows a summary of the main results.

---

[5]For iNat18 pretraining, we increase the number of prototypes to 8142, corresponding to the number of iNat18 classes.

Table 7: **SimCLR**: Evaluation of the mini-batch sampling distribution on various downstream task. Sampling imbalanced mini-batches lead to a significant drop in image classification tasks.

| | CIFAR100 | CIFAR100 1% | Place205 | Clevr/Count | Clevr/Dist | KITTI |
|---|---|---|---|---|---|---|
| 960 cls per batch | 69.9 | 31.4 | 52.1 | 77.4 | 65.5 | 70.5 |
| 8 cls per batch | 63.4 | 21.25 | 47.1 | 73.8 | 66.5 | 71.7 |
| Δ | -6.4 | -10.2 | -5.2 | -4.3 | +0.9 | +1.2 |

Table 8: **MSN**: Evaluation of the mini-batch sampling distribution on various downstream task. Sampling imbalanced mini-batches lead to a significant drop in image classification tasks.

| | CIFAR100 | CIFAR100 1% | Place205 | Clevr/Count | Clevr/Dist | KITTI |
|---|---|---|---|---|---|---|
| 960 cls per batch | 84.3 | 46.2 | 81.0 | 56.7 | 63.7 | 73.2 |
| 2 cls per batch | 71.4 | 26.4 | 76.4 | 50.3 | 65.6 | 71.5 |
| Δ | -12.9 | -13.2 | -4.6 | -6.4 | +1.9 | -1.6 |

Table 9: **VICReg**: Evaluation of the mini-batch sampling distribution on various downstream task. Sampling imbalanced mini-batches lead to a significant drop in image classification tasks.

| | CIFAR100 | CIFAR100 1% | Place205 | Clevr/Count | Clevr/Dist | KITTI |
|---|---|---|---|---|---|---|
| 960 cls per batch | 69.7 | 28.9 | 51.0 | 79.8 | 69.0 | 73.3 |
| 2 cls per batch | 60.8 | 16.9 | 44.9 | 76.8 | 69.84 | 72.1 |
| Δ | -8.9 | -12.0 | -6.0 | -3.0 | +0.7 | -1.1 |

Table 10: **data2vec**: Evaluation of the mini-batch sampling distribution on various downstream task. Sampling imbalanced mini-batches lead to a similar performances across tasks.

| | CIFAR100 | CIFAR100 1% | Place205 | Clevr/Count | Clevr/Dist | KITTI |
|---|---|---|---|---|---|---|
| 960 cls per batch | 50.3 | 13.7 | 37.0 | 76.8 | 49.7 | 65.3 |
| 2 cls per batch | 48.6 | 13.1 | 37.0 | 74.7 | 48.2 | 65.1 |
| Δ | -1.7 | -0.5 | 0 | -2.1 | +1.5 | -0.2 |

Table 11: **MAE**: Evaluation of the mini-batch sampling distribution on various downstream task. Sampling imbalanced mini-batches lead to a similar performance across tasks.

| | CIFAR100 | CIFAR100 1% | Place205 | Clevr/Count | Clevr/Dist | KITTI |
|---|---|---|---|---|---|---|
| 960 cls per batch | 75.0 | 28.3 | 50.4 | 90.4 | 72.4 | 70.0 |
| 2 cls per batch | 75.4 | 30.8 | 50.3 | 89.6 | 71.7 | 70.0 |
| Δ | +0.4 | +2.5 | -0.1 | -0.8 | -0.7 | +0.0 |

Table 12: **In-distribution:** Evaluation of the mini-batch sampling distribution on in-distribution ImageNet linear evaluation using 100% of the training set.

| | SimCLR | MSN | VICReg | data2vec | MAE |
|---|---|---|---|---|---|
| class balanced sampling | 66.9 | 77.1 | 69.1 | 41.5 | 65.9 |
| class imbalanced sampling | 55.8 | 59.4 | 51.4 | 40.6 | 65.8 |
| Δ | -11.1 | -17.7 | -17.7 | -0.8 | -0.1 |

Table 13: **In-distribution low-shot:** Evaluation of the mini-batch sampling distribution on in-distribution ImageNet linear evaluation using only 1% of the training set.

| | MSN | VICReg | data2vec | MAE |
|---|---|---|---|---|
| class balanced sampling | 66.2 | 48.6 | 27.4 | 35.1 |
| class imbalanced sampling | 28.0 | 18.1 | 31.4 | 34.8 |
| Δ | -38.2 | -30.5 | +4.0 | -0.3 |

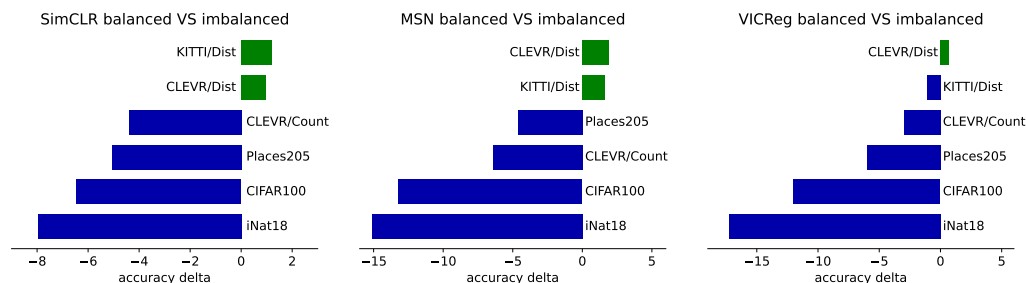

Figure 6: Visual representations of the results of Table 1. Methods relying on volume maximization regularizers all exhibit similar performance alteration across diverse transfer tasks.

