# OpenReview forum: "The hidden uniform cluster prior in self-supervised learning"
_ICLR.cc/2023/Conference — ICLR 2023 poster_

### Official Review · Reviewer_vagN · 2022-10-23

**Confidence:** 4
**Correctness:** 4
**Technical Novelty And Significance:** 2
**Empirical Novelty And Significance:** 4
**Recommendation:** 6

**Clarity, Quality, Novelty And Reproducibility:**

The paper is clear and somewhat novel in understanding the self-supervised representation learning. The empirical evaluation is extensive and convincing.

**Strength And Weaknesses:**

The  strengths of the paper:

+ The paper points out an interesting hidden prior used in the existing self-supervised representation learning methods and did extensive empirical evaluation to show the effects on down-stream classification task.

+ MSN is extended Prior Matching for Siamense Networks (PMSN) to incorporate arbitrary prior.


The weaknesses of the paper:

- While using a power-law distribution assumption to account for the long-tail class pre-training dataset leads to some improvements, it is somewhat marginal. However, it leads to significant performance degeneration when the pre-train dataset is class-balanced. In this sense, it potential value in application is limited.



**Summary Of The Paper:**


The paper reports the effects of a hidden prior (i.e., the uniform feature distribution assumption) used in the existing self-supervised pretraining methods for class-balanced datasets and real world class-imbalanced datasets. Moreover, a power-law distribution assumption to perfer long-tail prior is formulated for self-supervised pre-training and a method called Prior Matching for Siamse Networks (PMSN) is proposed. Extensive experiments are conducted on several benchmark datasets, showing that using the power-law distribution prior yields some improvements on class-imbalanced pretraining dataset (i.e. iNaturalist18) but leads to significant performance degeneration on class-balanced pre-training data (i.e., ImageNet).

**Summary Of The Review:**

The paper is clearly written and the experiments are extensive and convincing. The founding is novel and interesting. However, notice of that when using the long-tail class pre-training dataset and the power-law prior leads to marginal improvements, it potential value is limited.



=====
After reading the responses and the submission, the reviewer would like to increase the rating due to the contribution on implicit priors in self-supervised learning.

---

> ### Author Response · Authors · 2022-11-10
> **Response: R4**
>
> * Thank you for your constructive comment. We believe you raise a valuable point. And in fact, the main contribution of our paper is to highlight the important issue of implicit priors in self-supervised learning, and we believe this opens up lots of exciting future research directions (as supported by R1, R2, R3).
>   * As shown in section 3, these priors can be theoretically characterized.
>   * As shown in section 4, the uniform prior is not an entirely safe option and can lead to a large drop in performance when its assumptions are not satisfied, as shown in Table 1. This drop is indeed surprising and impressive, as indicated by Reviewer 1.
>   * And as shown in section 5 (see Figure 4) the uniform prior fails to capture semantic features with iNat18 pretraining, whereas the class-prior indeed captures semantic concepts. Additionally, as shown in Figure 3, the uniform prior can quite easily suppress relevant features if they do not satisfy its assumptions.
>
> We believe these contributions are a worthwhile and valuable addition to the community and researchers looking to design or leverage self-supervised learning algorithms. Besides downstream accuracy, uniform algorithmic priors have been linked to fairness issues of gender and racial bias in machine learning systems (Buolamwini, 2018; Bordia, 2019). We will include further discussion on this in the paper.

---

> > ### Author Response · Authors · 2022-11-17
> > **Response: R4 [Experiment Results]**
> >
> > - Re: weakness 1: After controlling for the model capacity in this experiment (as suggested by R3), we now find the delta increase from using a powerlaw prior on imbalanced data to be larger than the delta drop from using this prior on balanced data.
> >
> > We hope our new experiments and responses will be sufficient for you to consider increasing your score. In short, we have now:
> > 1. Theoretically identified and characterized implicit priors in popular self-supervised methods.
> > 2. Empirically demonstrated the potential detrimental effect of these implicit priors both qualitatively and quantitatively when pretraining with class-imbalance data.
> > 3. Demonstrated how changing this prior can lead the model to capture different (and more semantic) sets of features, and observed that using a mixture of priors can enable the model to capture multiple feature sets following various distributions (including the default uniform distributed features).
> >
> > We believe this analysis is of significant value to the machine learning community, and should help inform future design and application of self-supervised learning methods.

---

### Official Review · Reviewer_sHjw · 2022-10-24

**Confidence:** 4
**Correctness:** 3
**Technical Novelty And Significance:** 3
**Empirical Novelty And Significance:** 3
**Recommendation:** 6

**Clarity, Quality, Novelty And Reproducibility:**

Clarity: 7
Quality: 6
Originality: 7
Reproducibility: the source code isn't provided. The main algorithmic change is in Eq. 9, which seems straightforward to implement.

**Strength And Weaknesses:**

Strength:
1. It is probably the first paper in SSL domain to classify and formulate the current typical SSL methods into more or less constrained k-means problems with detailed proofs. which can be referred to as a review from the theoretical perspective.
2. The paper utilizes smartly designed experiments to demonstrate the negative impact of prior mismatch (in mini-batch) to SSL with volume maximization regularizers.
3. Performance improvement of pretraining on class-imbalanced dataset is consistent on multiple classification downstream tasks.

Weakness:
1. Unfair comparison with different models on different pretext dataset. In Table 3. The model pretrained on iNat18 is ViT-S while the one on ImageNet-1K is ViT-B. The capacities of this two model are very different (the latter is 4 times larger than former i.t.o # parameters) compared to the size differences of the two dataset (roughly 2 times). To better demonstrate the gain of power-law prior on the class-imbalanced dateset, the model impact should be controlled.
2. Limited gain on choosing long-tailed prior over uniform. As shown in Table 2 and Table 3, the performance improvement on classification is minor compared to the loss with power-law prior on balanced dataset. Looks like without the knowledge of class distribution, the uniform prior is a much safer option.

**Summary Of The Paper:**

This paper points out that the overlooked uniform class prior would harm the learning of semantic representation for SSL methods, especially those with collapse prevention techniques, on class-imbalanced real-world datasets. The paper formulates and proves the objectives of the recent representative SSL methods to be essentially k-means problems with assumption of uniform prior. The paper shows the negative impact of uniform prior on class-imbalanced dataset through a comparative experiment on sampling strategies and visualizes the learned prototypes. The paper subsequently proposes a remedy by extending the objective of MSN through the recovering of the entropy term back to the relative entropy from an arbitrary prior to the average prediction. The arbitrary prior to the paper’s interest is the power-law distribution which is often close to a typical real-world dataset. Alongside with a toy experiments to showcase the impact of different priors and mini-batch distributions, downstream tasks of classification and object distance detection are compared under different pretraining datasets and priors and shows the performance gain on class-imbalanced dataset with power-law prior. With different impacts of prior mismatch on uniform and non-uniform datasets, the author notes that the class-imbalanced pretraining issue is not completely solved.

**Summary Of The Review:**

The paper formulates current SSL as K-means problem and points out the overlooked uniform prior in those methods with volume maximization techniques and demonstrate that it can harm the model pretrained on class-imbalanced datasets, which is usually the case in real world. The proposed method is a straightforward extension of MSN. With power-law prior, it yields improved performance over uniform prior on class-imbalanced dataset. However, experiments in this paper also shows that the cost of the prior mismatch over balanced
dataset is much more higher than over class-imbalanced dataset, which suggests the uniform prior is a safer prior when there is no access to the label. Furthermore, there is a potential of unfair comparison in the experiment reported Table 3.

On the whole, my recommendation of the paper is marginal accept

---

> ### Author Response · Authors · 2022-11-10
> **Response: R3**
>
> Indeed, one of the main contributions of the paper is to highlight and theoretically characterize the implicit priors in self-supervised learning, as you have summarized in your review, and as also supported by reviewers R1, R2, and R4.
> * Re. weakness 1: Yes, this is a good point. We will train an identical model on IN1k and will report back here once we have the results.
> * Re. weakness 2: In fact, the uniform prior is not an entirely safe option and can lead to a large drop in performance when its assumptions are not satisfied, as shown in Table 1. This drop is indeed surprising and impressive, as indicated by Reviewer 1. Moreover, the qualitative analysis in Figure 4 shows that the uniform prior fails to capture semantic features with iNat18 pretraining, whereas the class-prior indeed captures semantic concepts. Additionally, as shown in Figure 3, the uniform prior can quite easily suppress relevant features if they do not satisfy its assumptions. However, you do raise an important point. We believe our paper highlights an important issue for the community and opens up lots of exciting future research directions around priors in self-supervised learning. Besides downstream accuracy, uniform algorithmic priors have been linked to fairness issues of gender and racial bias in machine learning systems (Buolamwini, 2018; Bordia, 2019). We will include further discussion on this in the paper.

---

> > ### Author Response · Authors · 2022-11-17
> > **Response: R3 [Experiment Results]**
> >
> > - Re: experiment 1 (capacity of models pretraining on iNat18 vs IN1k): We trained a similar capacity model (ViT-S/16) on IN1k with and without a powerlaw prior, and found the same conclusions to hold. In fact, the delta increase from using a powerlaw prior on imbalanced data is now larger than the delta drop from using this prior on balanced data. We thank the reviewer for this suggestion of controlling for model impact, as it has now providing further compelling evidence for the advantage of long-tailed (e.g., powerlaw) priors. We will update the paper with this result.

---

### Official Review · Reviewer_1J6n · 2022-10-25

**Confidence:** 4
**Correctness:** 4
**Technical Novelty And Significance:** 3
**Empirical Novelty And Significance:** 3
**Recommendation:** 6

**Clarity, Quality, Novelty And Reproducibility:**

- The paper is well written with clear demonstration of the formulations, observations and solutions.
- The quality and the novelty of this paper are good overall.
- I'm confident to reproduce the paper's proposed algorithm.

**Strength And Weaknesses:**

Strength:
- Class imbalanced data is ubiquitous and deserves attention, especially for self-supervised setting where you don't know the data distribution.

Weakness:
- The only disappointment I had is that the authors only demonstrated the performance comparison for the MSN algorithm, where less deviation is needed. It would be more comprehensive if the authors can show how long tail distribution prior can be enforced in other algorithms.
- The experiment is clear and convincing, but it seems only image modality is considered. I'm not sure how much this affects the generalizability of the problem/solution.
- It would also be a good sanity check to visualize the features learned from class-balanced sampling vs class-imbalanced sampling.

**Summary Of The Paper:**

This paper points out a phenomenon that typical self-supervised learning losses implicitly enforces a feature prior that forms clusters of roughly equal size. The authors empirically verifies that this behavior can hurt the performance if the data samples are class-imbalanced. Then the authors proposed a variant that assumes other prior class distribution (i.e. long tail distributions such as power-law), and demonstrated that this could remedy the problem.

**Summary Of The Review:**

I like the simplicity and the insight of the paper. The only concern is that only one algorithm and its variant is demonstrated.

---

> ### Author Response · Authors · 2022-11-10
> **Response: R2**
>
> * Regarding the focus only on MSN: Thank you for this constructive comment. We will run experiments changing the prior with the SwAV method as well, and will report back here when we have the results. In general, while adapting the prior is unclear with certain methods for which this bias is deeply implicit, it should be more straightforward to adapt the prior with prototype based methods (SeLA, DINO, DeepCluster, etc.).
> * Regarding the focus on images: It is true, the focus of this paper is on the image modality, and all methods considered are state-of-the-art methods for self-supervised learning from images. We will make this clear in the introduction. We believe the insights we highlight are likely to hold in other modalities as well, but we consider it beyond the scope of this present work. It is however a great suggestion for future work to look into.
> * Regarding visualizing features learned with various sampling strategies, we fully agree with this point, and in fact Figure 2 in the paper visualizes the features learned from class-balanced sampling vs class-imbalanced. We find the emergent features when pretraining with class-balanced sampling to be associated with high level concepts, such as specific ImageNet classes. By contrast the features obtained with class-imbalanced sampling tend to be associated with low-level concepts, such as shape, pose, or texture.

---

> > ### Author Response · Authors · 2022-11-17
> > **Response: R2 [Experiment results]**
> >
> > - Re. experiment 2 (powerlaw prior with another method): This suggested experiment also worked, and we thank the reviewer for this recommendation. By changing the sinkhorn-normalization distribution in the widely popular SwAV method to a powerlaw distribution, when pretraining with a ResNet50, we obtain very similar observations as with MSN pretrained with a vision transformer. Simply pretraining on iNat18 (long-tailed distribution) with this powerlaw prior directly leads to a similar improvements over the default uniform prior on all the semantic downstream tasks. We will also update the paper with this result.

---

### Official Review · Reviewer_JLgU · 2022-10-28

**Confidence:** 2
**Correctness:** 3
**Technical Novelty And Significance:** 3
**Empirical Novelty And Significance:** 3
**Recommendation:** 6

**Clarity, Quality, Novelty And Reproducibility:**

This paper is well written and easy to follow. Details about experiment settings can be found in the paper.

**Strength And Weaknesses:**

Strength:
1. The analysis of SSL methods with volume maximization principle on class-imbalanced data is novel and impressive
2. Experiment analysis of the performance degeneration on class-imbalanced data is thorough and impressive
3. The findings of the connection between SSL methods and k-means is insightful

Weakness:
1. The toy experiment can be further analyzed. From Figure 3, the power-law prior encourages the model to retain semantic features related to MNIST but discard semantic features related to CIFAR10 (the nearest neighbors have different CIFAR classes). It seems that the choice of prior decides which types of class conditional semantic features will be retained (uniform or power-law). Is it possible to retain semantic features for both MNIST and CIFAR (e.g., by using a mixture prior?). Hope to see more analysis.

2. The power-law prior is only tested on MSN. As several volume maximization based SSL methods are discussed in the paper, a natural question is that whether introducing the power-law prior will be helpful for these method.


**Summary Of The Paper:**

This paper formulates several self supervised learning methods (VICReg, SimCLR, MSN, SwAV) with the volume maximization principle as variants of k-means, analyzes their difference on class-balanced and class-imbalanced data, and indicates that the uniform prior w.r.t. prototype vectors is the curse of performance degeneration of class-imbalanced data. For resolving this problem, this paper develops PMSN, an extension to MSN which replaces the uniform prior with a power-law distribution. Experiments verifies the effectiveness of the proposed method.

**Summary Of The Review:**

In general, this paper focuses on an important and interesting problem w.r.t. self supervised learning, and its findings are insightful. Experiments can be further improved to better support the claims and verifies theoretical analysis made in the paper.

---

> ### Author Response · Authors · 2022-11-10
> **Response: R1**
>
> Indeed, one of the main contributions of this work is to show that the implicit prior in self-supervised methods affects which features will be retained, and by consciously changing this prior one can guide the model to capture different features.
> * Re. weakness 1: Indeed it is a very interesting question whether one can retain multiple sets of features with different distributions. We will experiment with the proposed mixture prior in this setting and report back here when we have the results.
> * Re. weakness 2: Thank you for the constructive suggestion. We are running experiments changing the prior with the SwAV method as well, and will report  back here when we have the results. In general, while adapting the prior is unclear with certain methods for which this bias is deeply implicit, it should be more straightforward to adapt the prior with prototype based methods (SeLA, DINO, DeepCluster, etc.).

---

> > ### Author Response · Authors · 2022-11-17
> > **Response: R1 [Experiment results]**
> >
> > - Re. experiment 1 (mixture prior): Remarkably, the suggested experiment of training with a mixture of priors worked, and we  thank the reviewer for this suggestion. Indeed, we were able to capture both the digit information (powerlaw distributed) and the semantic features (uniform distributed) with this mixture prior! In particular, we trained the model with two heads. For one head, we enforced a uniform prior, and for the other we enforced a powerlaw prior. The total loss was the sum of the losses from the two heads. We will update the paper with this mixture result.
> >
> > - Re. experiment 2 (powerlaw prior with another method): We were successfully able to observe the same trends by changing the prior in the widely popular SwAV method, and thank the reviewer for this suggestion. By changing the sinkhorn-normalization distribution in SwAV to a powerlaw distribution, when pretraining with a ResNet50, we obtain very similar observations as with MSN pretrained with a Vision Transformer. Simply pretraining on iNat18 (long-tailed distribution) with this powerlaw prior directly leads to similar improvements over the default uniform prior on all semantic downstream tasks. We will also update the paper with this result.

---

### Author Response · Authors · 2022-11-10
**Review Summary**

We thank the reviewers for their consideration in providing thoughtful reviews of our paper. As summarized in the reviews, this paper:
* provides a novel and insightful empirical and theoretical characterization of the hidden prior in existing self-supervised representation learning methods (Reviewers 1, 2, 3, 4)
* conducts surprising, clear, and convincing experiments showcasing the effect of this prior when its assumptions are not satisfied; i.e., with class-imbalanced data (Reviewers 1,2,3)

We are running some of the experiments suggested by reviewers and will report back once we have these results.

---

> ### Author Response · Authors · 2022-11-17
> **Experiment Results**
>
> We have run the suggested experiments, and with thanks to the reviewers, have now further strengthened the findings of our work. In short, new experiments show that:
> 1)  Using a mixture of priors in self-supervised learning can enable the model to capture multiple feature sets following various distributions (including the uniform distributed features preferred implicitly by numerous popular SSL methods).
> 2) When controlling for model capacity, the drop in performance when the assumptions of the default uniform prior are not satisfied (e.g., with any class-imbalanced data) is greater than the drop when pretraining on class-balanced data with a long-tailed prior, and these conclusions also hold for downstream transfer tasks.
> 3) Changing the prior in another SSL method (SwAV) can lead to similar empirical improvements in performance when pretraining with naturally class-imbalanced data, even when using an entirely different model architecture (ResNet instead of Vision Transformer).
>
> Once again, we are very grateful to the reviewers for their suggestions. We believe the novel analysis provided in this work is of significant value to the machine learning community, and should help inform future design and application of self-supervised learning methods. In particular, our work sheds light on implicit priors in self-supervised learning, and the effect of these priors on the features captured by the model during pretraining.

---

### Public Comment · ~Zhihan_Zhou2 · 2023-02-13
**Congratulations to the nice work! Recommend our work on self-supervised long-tailed learning.**

Hi,

Congratulations to the nice work! Our recent ICML'22 paper BCL (https://proceedings.mlr.press/v162/zhou22l/zhou22l.pdf) also studied self-supervised long-tailed learning. Would you mind citing our work? Many thanks!

Best,

Zhihan

---

> ### Author Response · Authors · 2023-02-20
> **Thank you!**
>
> Hi Zhihan,
>
> Thank you for pointing out your interesting work on the topic! Yes would be happy to include a reference.
>
> Best,
> Mido

---

### Decision · Program_Chairs · 2023-01-20

**Decision:**

Accept: poster

**Justification For Why Not Higher Score:**

Although all the reviewers were on the positive side, there were no strong accept scores for the paper.  The suggested experiments made by the reviewers appear to have been carried out by the authors but some details on these experiments were not given and it's unclear how much of this will go into the final paper.  I encourage the authors to include these experiments, at least in an appendix.

**Justification For Why Not Lower Score:**

With all four reviewers at least somewhat positive about the paper, particularly about its importance and novelty, I think the paper warrants being accepted at the conference.

**Metareview: Summary, Strengths And Weaknesses:**

Thanks for your submission to ICLR.

All four reviewers ultimately leaned toward accept on this paper, so there is some consensus that the paper is suitable for publication at ICLR.  On the strengths, reviewers noted the importance of the problem, the novelty of the proposed solution, and noted strength in some of the experimentation.  On the negative side, several reviewers noted some additional experimentation that would improve the paper.  During the rebuttal and discussion period, the authors responded with some additional results based on the feedback, and at least one reviewer raised their score.  In the end it seems everyone is reasonably positive about the paper.

**Note From Pc:**

if the above contains the word "oral" or "spotlight" please see: "oral" presentation means -> notable-top-5% and "spotlight" means -> notable-top-25%. As stated in our emails, we are disassociating presentation type from AC recommendations